# Unraveling the genetic links between stature and disease in East Asians: A multi-biobank genetic correlation and risk prediction study

Ying-Ju Lin[1,2⊕], Ting-Yuan Liu[3⊕], Jai-Sing Yang[4], Ju-Pi Li[5,6], Jian-Shiun Chiou[7,8], Hsing-Fang Lu[3,9], Kuyuri Ariyoshi[9,10,11], Keiko Hikino[12], Chikashi Terao[9,10,13], Chen-Hsing Chou[7,8], Wen-Miin Liang[8], I-Ching Chou[2,14], Ting-Hsu Lin[1], Chiu-Chu Liao[1], Shao-Mei Huang[1], Fuu-Jen Tsai[1,2,15,16]*

1 Genetic Center, Department of Medical Research, China Medical University Hospital, Taichung, Taiwan, 2 School of Chinese Medicine, College of Chinese Medicine, China Medical University, Taichung, Taiwan, 3 Million-Person Precision Medicine Initiative, Department of Medical Research, China Medical University Hospital, Taichung, Taiwan, 4 Department of Medical Research, China Medical University Hospital, China Medical University, Taichung, Taiwan, 5 Department of Pathology, School of Medicine, Chung Shan Medical University, Taichung, Taiwan, 6 Department of Pediatrics, Chung Shan Medical University Hospital, Taichung, Taiwan, 7 PhD Program for Health Science and Industry, College of Health Care, China Medical University, Taichung, Taiwan, 8 Department of Health Services Administration, China Medical University, Taichung, Taiwan, 9 Laboratory for Statistical and Translational Genetics, RIKEN Center for Integrative Medical Sciences, Yokohama, Japan, 10 Clinical Research Center, Shizuoka General Hospital, Shizuoka, Japan, 11 Division of Medical Biophysics, Department of Biophysics, Graduate School of Health Sciences, Kobe University, Kobe, Japan, 12 Laboratory for Pharmacogenomics, RIKEN Center for Integrative Medical Sciences, Yokohama, Japan, 13 The Department of Applied Genetics, The School of Pharmaceutical Sciences, University of Shizuoka, Shizuoka, Japan, 14 Division of Pediatrics Neurology, China Medical University Children's Hospital, Taichung, Taiwan, 15 Division of Medical Genetics, China Medical University Children's Hospital, Taichung, Taiwan, 16 Department of Medical Laboratory Science & Biotechnology, Asia University, Taichung, Taiwan

⊕ These authors contributed equally to this work.
* 000704@tool.caaumed.org.tw

## Abstract

Both genetic and environmental factors affect human stature, including overall height and familial short stature (FSS), and it is associated with various health outcomes. However, the study of genetic connections between stature and health conditions remains lacking in East Asian populations. Hence, we conducted parallel genome-wide association studies (GWAS) of body height and FSS in the Han Taiwanese population, aiming to elucidate the genetic influences of stature on health and facilitate the formulation of precision-health strategies. We analyzed large-scale GWAS data on adult height (120,301 Han Taiwanese) and FSS (FSS; 2,050 cases, 27,966 controls) to examine cross-trait genetic correlations across five East Asian biobanks, and applied phenome-wide association studies (PheWAS) and polygenic risk score (PRS) analyses to assess clinical outcomes using Cox proportional hazard models and Kaplan–Meier analyses. We identified 293 loci for height and five for FSS, with cross-biobank genetic correlations linking stature to body size, lung function, and

**Data availability statement:** We conducted genome-wide association study (GWAS) analyses to obtain body height summary statistics from the Taiwan Biobank and FSS summary statistics from the China Medical University Hospital (CMUH) Biobank. Additional GWAS summary statistics of East Asian ancestry were sourced from TWB (https://pheweb.twbiobank.org.tw:5038/about), BioBank Japan (BBJ) (https://pheweb.jp/), Korean Genome and Epidemiology Study (KoGES) (https://koges.leelabsg.org/), China Kadoorie Biobank (CKB) (https://pheweb.ckbiobank.org/), and CMUH Biobank (http://cmuh-biobank.eastasia.cloudapp.azure.com/). TWB, BBJ, KoGES, CKB, and CMUH data are available online.

**Funding:** We thank the China Medical University, Taiwan (CMU111-MF-21 to YJL, CMU111-S-35 to YJL, and CMU113-MF-47 to YJL), the China Medical University Hospital, Taiwan (DMR-112-042 to FJT, DMR-113-038 to FJT, DMR-113-103 to YJL, DMR-114-015 to FJT, DMR-114-088 to YJL, DMR-115-044 to FJT, and DMR-115-120 to YJL), and the Ministry of Science and Technology, Taiwan (NSTC 113-2320-B-039-041 to YJL, NSTC 114-2314-B-039-019 to ICC, NSTC 114-2314-B-039-020 to FJT, NSTC 114-2320-B-039-046 to YJL, NSTC 114-2813-C-039-060-B to YJL, and NSTC 114-2813-C-039-147-B to FJT) for supporting this study. The funders had no role in study design, data collection and analysis, decision to publish, or preparation of the manuscript.

**Competing interests:** The authors have declared that no competing interests exist.

cardiovascular/reproductive traits (atrial flutter/fibrillation [AF], menarche, and endometriosis). PheWAS showed that height PRS increased risks of AF and endometriosis, while FSS PRS had a protective effect against endometriosis. MR analyses showed that taller stature increased AF risk independently and endometriosis risk through menarche/weight, while shorter stature had a weak protective effect against endometriosis. Survival analyses showed the association of higher height PRS with greater AF risk and an earlier divergence of cumulative incidence curves. These time-to-event patterns were consistently replicated using meta-analysis–derived PRSs. The findings highlight stature-related genetic determinants, associated health outcomes, and polygenic risk scores as effective tools for early risk prediction and precision health strategies in East Asian populations.

## Author summary

Adult stature, including height and familial short stature (FSS), is influenced by genetic and environmental factors. Although stature has been associated with various health conditions, the genetic basis of stature-related health risks remains underexplored, especially in East Asian populations. Herein, genome-wide association studies (GWAS) of >120,000 Han Taiwanese individuals were analyzed, and 293 loci for height and 5 for FSS were identified. Cross-trait genetic correlations were assessed across five East Asian biobanks and analyzed via phenome-wide association studies and polygenic risk score (PRS) to evaluate clinical implications. Stature traits showed strong genetic correlations with body size, lung function, and cardiovascular and reproductive conditions, particularly AF, menarche timing, and endometriosis. Tall stature was associated with increased risks of AF and endometriosis, whereas short stature conferred a weak protective effect against endometriosis. These associations, confirmed through PRS-based risk estimates and divergence in survival curves, support the use of stature-related genetics for improved disease risk stratification. Our findings indicate that stature is a genetically informed, non-modifiable risk factor with potential utility in early risk prediction and precision health for East Asian populations.

## Introduction

Height is a fundamental aspect of pubertal growth, shaped by both genetic and environmental factors [1,2]. Final stature is determined by the maturation of growth plates—typically concluding around puberty under the regulation of hormones and growth factors [3,4]. Large genome-wide association studies (GWAS) in European cohorts have identified thousands of height-associated loci [5–7], some of which correlate with health outcomes. Polygenic risk scores (PRS) for adult height have been demonstrated to be as predictive as mid-parental height for identifying children with lower expected height, while genetically determined taller stature is linked to lower

low-density lipoprotein-cholesterol (LDL-C) levels, leaner body composition, and larger overall body size [8,9]. Furthermore, longitudinal research indicates that childhood growth patterns are associated with future risks of cardiovascular disease and type 2 diabetes—which often manifest or impact health in later life [10,11].

Familial (genetic) short stature (FSS)—the most common form of short stature [12–14]—is defined as a final height below the third percentile, with normal bone age, puberty onset, and growth rate, coupled with a family history of short stature, while excluding dysmorphisms, abnormal puberty, or thyroid dysfunction [15,16]. PRS for FSS consistently indicate lower height trajectories, and several FSS-associated variants overlap with genes involved in growth plate development, the growth hormone–insulin-like growth factor-1 axis, and established height GWAS loci [17,18]. Notably, these genes influence skeletal growth, oxidative metabolism, and biological pathways that also link early-life growth patterns with health outcomes [19,20].

Studies have shown that genetic loci associated with body height and FSS are related to a broad range of health outcomes [8–11,19–22]; however, these relationships with disease traits remain insufficiently characterized in East Asian populations. To address this gap, we conducted parallel GWAS of body height and FSS in a large Han Taiwanese cohort. The resulting GWAS summary statistics were used for cross-trait genetic correlation analyses with data from five major East Asian biobanks. This strategy allowed for the identification of key diseases and traits that share a common genetic architecture with body height and FSS. Additionally, this study aimed to clarify the broader health implications of stature-associated genetic variants to support the development of precision health strategies tailored to East Asian populations.

## Results

### Study overview

We performed parallel GWAS on height (120,301 Han Taiwanese from Taiwan Biobank [TWB]) and FSS (FSS; 2,050 cases and 27,966 controls from China Medical University Hospital [CMUH]) (Fig 1), followed by extended cross-trait genetic association analyses across five East Asian biobanks—TWB (n = 145,490), CMUH (n = 318,516), China Kadoorie Biobank (CKB) (n = 512,000), BioBank Japan (BBJ) (n = 178,726), and Korean Genome and Epidemiology Study (KoGES) (n = 72,298)—encompassing individuals of Han Taiwanese, Han Chinese, Japanese, and Korean ancestries. Significant genetic correlations were classified into height-specific traits (e.g., weight, hip and waist circumference, height, expiratory reserve volume, inspiratory capacity, atrial fibrillation [AF], menarche, creatinine, drinking habits, white blood cell count, neutrophil count, platelet count, ATC_C10AA, and LDL-C), shared traits (e.g., weight, hip and waist circumference, height, expiratory reserve volume, inspiratory capacity), and FSS-specific traits (e.g., body mass index [BMI] and endometriosis).

### Body height genetic loci and health conditions

We conducted a GWAS for body height using 120,301 participants (S1 Fig). In the [TWB5] HEIGHT GWAS study, the 120,301 participants had a mean age of 49.55 years (SD = 11.14) and were predominantly female (62.5%) (S1 Table). The average body height was 162.24 cm (SD = 8.32). We identified 293 lead single-nucleotide polymorphisms (SNPs) across distinct loci (Fig 2A and S2–S4 Tables).

S5 and S6 Tables showed 1,305 body height-related genes through SNP-based annotation and gene-based GWAS analysis using Functional Mapping and Annotation of GWAS and Multi-marker Analysis of GenoMic Annotation (MAGMA) [23,24] via [TWB5]HEIGHT GWAS summary statistics. SNP2GENE in FUMA (https://fuma.ctglab.nl/snp2gene) highlighted 1,185 genes—most strongly *LCORL*, *PAN2*, and *CNPY2* (minGwas$P \leq 2.4 \times 10^{-130}$)—with additional multi-population support for *RP11-977G19.10*, *CS*, *IL23A*, and *DCAF16* (S5 Table). MAGMA further identified 619 genome-wide significant genes ($P < 0.05$/18,651), including *ADAMTS17, FNDC3B, RAD51B, IGF1R*, and *DLEU1* (S6 Table and S2 Fig).

We examined the genetic architecture of stature in Han Taiwanese individuals through a GWAS of body height in 120,301 participants, identifying 1,305 genes—1,185 from SNP-based annotations and 619 from gene-based analyses—across 293 distinct loci (S2–S6 Tables). We implemented a rigorous three-stage verification framework by integrating

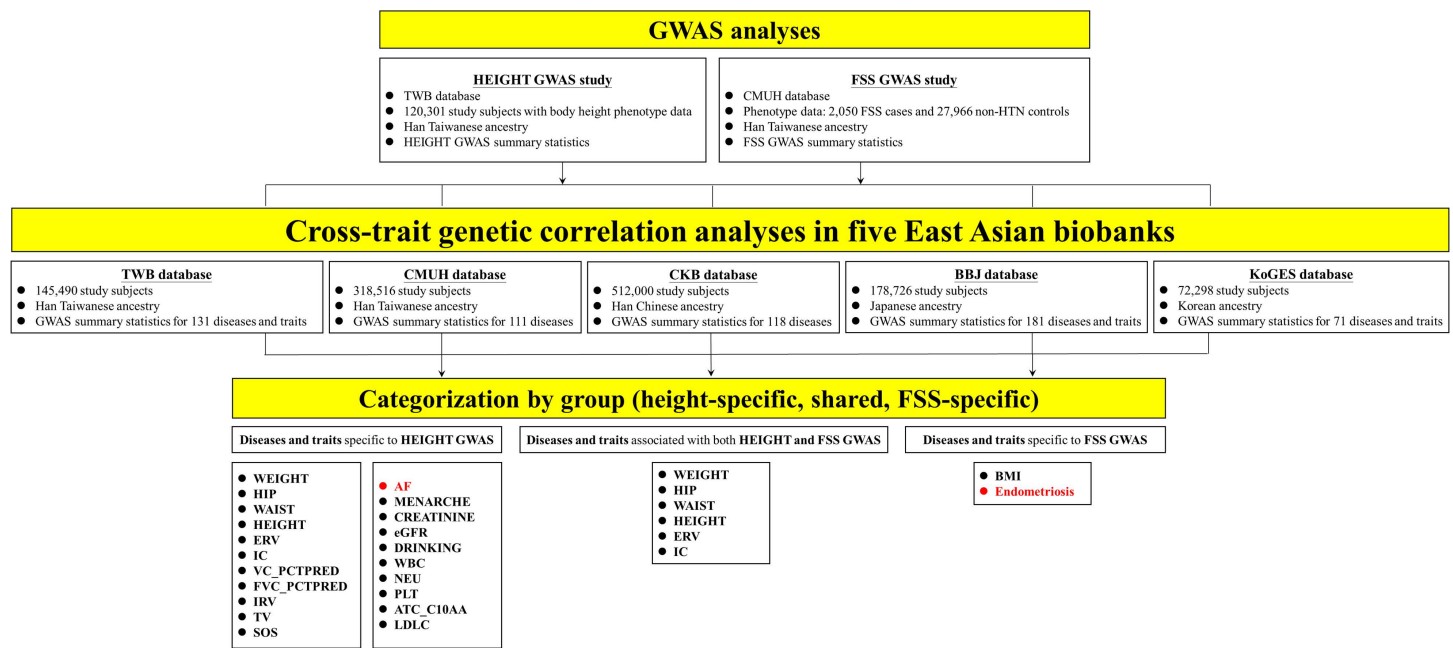

**Fig 1. Study overview.** GWAS of height (120,301 Han Taiwanese from TWB) and FSS (2,050 cases and 27,966 controls from CMUH) were conducted, followed by cross-trait genetic correlation analyses across five East Asian biobanks (TWB, CMUH, CKB, BBJ, KoGES). Significant genetic correlations were classified into height-specific, FSS-specific, or shared traits. Abbreviations: AF, atrial flutter/fibrillation; ATC_C10AA, HMG-CoA reductase inhibitors (Anatomical Therapeutic Chemical code C10AA); BBJ, Biobank Japan; BMI, body mass index; CKB, China Kadoorie Biobank; CMUH, China Medical University Hospital; CREATININE, serum creatinine; DRINKING, ≥ 6 months of alcohol consumption; eGFR, estimated glomerular filtration rate; ERV, expiratory reserve volume; FSS, functional somatic syndrome; FVC_PCTPRED, % of predicted forced vital capacity; GWAS, genome-wide association study; HEIGHT, body height; HIP, hip circumference; HTN, hypertension; IC, inspiratory capacity; IRV, inspiratory reserve volume; KoGES, Korean Genome and Epidemiology Study; LDLC, low-density lipoprotein cholesterol; MENARCHE, age at first menstruation; NEU, neutrophil count; PLT, platelet count; SOS, speed of sound (QUS); TV, tidal volume; TWB, Taiwan Biobank; VC_PCTPRED, % of predicted vital capacity; WAIST, waist circumference; WBC, white blood cell count; WEIGHT, body weight.

internal GCTA-COJO joint analysis, external conditional adjustments against a comprehensive multi-ancestry reference catalog, and physical LD profiling (±10 Mb) to distinguish novel associations from established signals. Loci were strictly defined as novel if they retained genome-wide significance after conditioning on known variants ($P_{cond} < 5 \times 10^{-8}$) and exhibited negligible linkage disequilibrium ($r^2 < 0.1$) with established signals, distinguishing them from independent secondary signals ($0.1 \leq r^2 < 0.6$) and known associations ($r^2 \geq 0.6$ or statistically explained). Using these criteria, 16 novel height-associated loci mapping to *GIGYF2, MAP3K13, NSD2, THBS2, EXTL3, ZFAT, FBP1, LTBP3, SPRY2, SLC12A6, PIEZO1, LNCNEF, SNX21*, and three uncharacterized regions (*AC097637.3, AC019131.1*, and *AC018645.2*) were identified (S3 Table).

We performed Bayesian colocalization analysis using the coloc R package (https://cran.r-project.org/web/packages/coloc/vignettes/a01_intro.html) to rigorously assess the consistency of association signals across ancestries [25]. This approach facilitated a probabilistic determination of whether overlapping signals represent shared (posterior probability H4 (PP.H4.abf)) or distinct (PP.H3.abf) causal variants. Analysis of the 293 height-associated genomic regions revealed substantial genetic homogeneity between the Taiwan Biobank (TWB) and other East Asian populations (S4 Table); 56–72% of loci showed strong evidence for a shared causal variant (posterior probability H4 (PP.H4.abf) > 0.7) compared to KoGES, BBJ, and datasets in Yengo et al. (EAS) [5,20,26]. Conversely, comparisons with European cohorts (EUR and Yengo et al. EUR) demonstrated greater genetic divergence, with approximately 40% of loci supporting the hypothesis of distinct causal mechanisms (PP.H3.abf > 0.7).

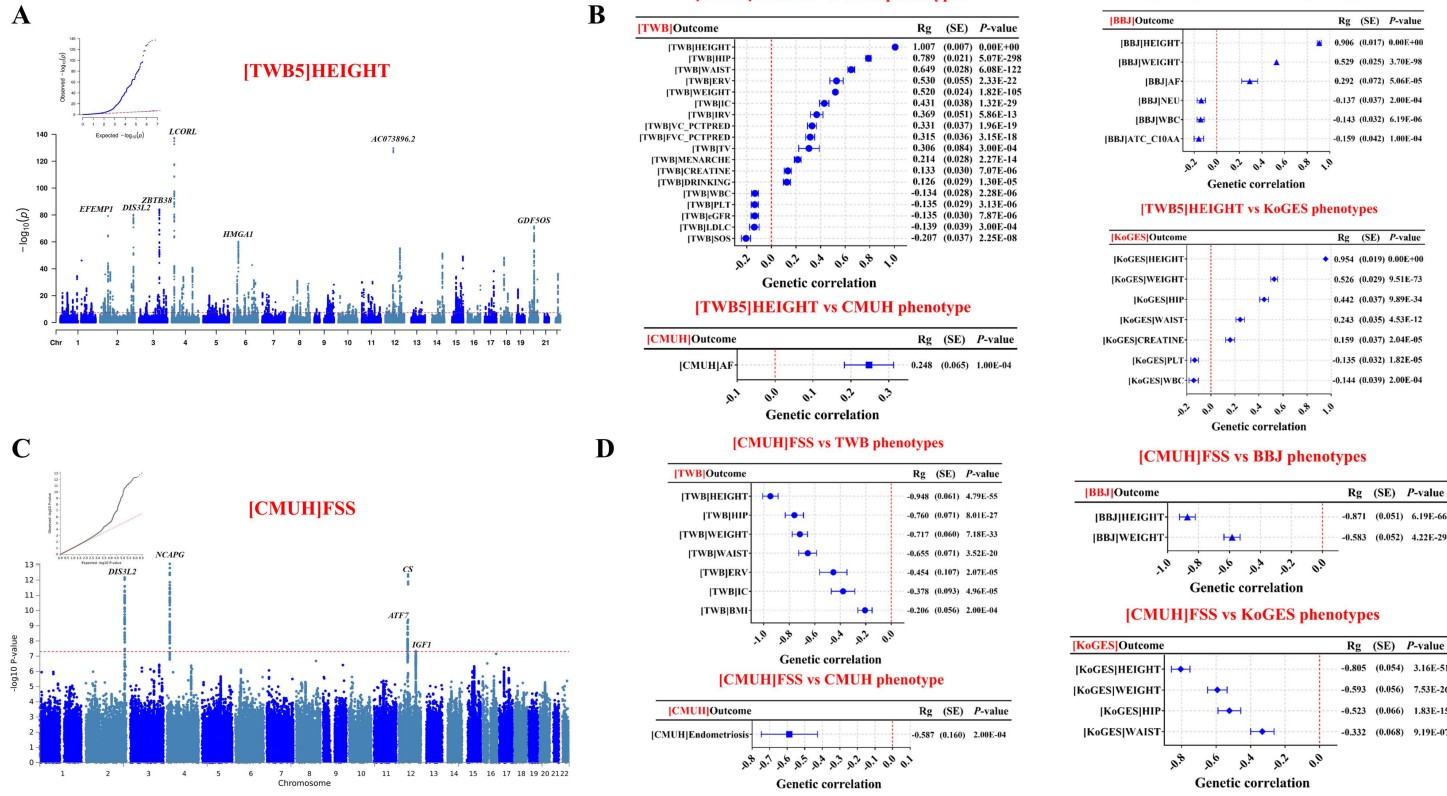

**Fig 2. Genome-wide association studies (GWAS) and genetic correlations for body height and FSS in East Asians.** A. Manhattan and QQ plots for the body height GWAS of 120,301 Han Taiwanese in TWB, showing $-\log_{10}(P)$ across chromosomes ($P < 5 \times 10^{-8}$). B. Genetic correlations (Rg, standard error, and P-value) for [TWB]HEIGHT with key phenotypes in East Asian biobanks: 18 in TWB, one in CMUH, six in BBJ, and seven in KoGES. C. Manhattan and QQ plots for the FSS GWAS of 2,050 FSS cases and 27,966 controls in CMUH ($P < 5 \times 10^{-8}$). D. Genetic correlations (Rg, standard error, and P-value) for [CMUH]FSS with phenotypes in East Asian biobanks: seven in TWB, one in CMUH, two in BBJ, and four in KoGES. Abbreviations: AF, atrial flutter/fibrillation; ATC_C10AA, HMG-CoA reductase inhibitors (Anatomical Therapeutic Chemical code C10AA); BBJ, Biobank Japan; BMI, body mass index; CMUH, China Medical University Hospital; CKB, China Kadoorie Biobank; CREATININE, serum creatinine; DRINKING, ≥6 months of alcohol consumption; eGFR, estimated glomerular filtration rate; ERV, expiratory reserve volume; FSS, familial short stature; FVC_PCTPRED, % of predicted forced vital capacity; HEIGHT, body height; HIP, hip circumference; IC, inspiratory capacity; IRV, inspiratory reserve volume; KoGES, Korean Genome and Epidemiology Study; LDLC, low-density lipoprotein cholesterol; LDSC, Linkage Disequilibrium Score Regression; MENARCHE, age at first menstruation; NEU, neutrophil count; PLT, platelet count; SE, standard error; SOS, speed of sound (QUS); TV, tidal volume; TWB, Taiwan Biobank; VC_PCTPRED, % of predicted vital capacity; WAIST, waist circumference; WBC, white blood cell count; WEIGHT, body weight.

Cross-trait genetic correlation analyses revealed significant associations between height and multiple traits across different cohorts—18 in TWB, 1 in CMUH, none in CKB, 6 in BBJ, and 7 in KoGES—after multiple testing correction using the Bonferroni method (Fig 2B and S7-S11 Tables). A higher genetic predisposition to taller stature was associated with larger body size, enhanced lung function, delayed menarche, lower LDL-C levels, reduced kidney function, and a heightened AF risk, highlighting significant genetic implications for health.

## FSS genetic loci and health conditions

We identified 2,050 FSS cases (mean age 11.1±5.4 years; 50.0% male) and 27,966 controls (mean age 44.4±15.1 years; 45.4% male), with significant age and sex differences ($P < 0.001$) (S12 Table). In a GWAS of 2,050 FSS cases and 27,966 controls of Taiwanese ancestry adjusting for age, sex, and 10 PCs, we identified five significant FSS-associated loci—*DIS3L2*, *NCAPG*, *ATF7*, *CS*, and *IGF1* (Fig 2C and S13–S15 Tables).

S16 and S17 Tables identified 27 FSS-related genes from [CMUH]FSS GWAS summary statistics using two complementary approaches: SNP-based annotation via FUMA SNP2GENE and gene-based GWAS analysis via MAGMA [23,24]. SNP-based annotation highlighted 26 genes, with the strongest signals at *DCAF16, NCAPG,* and *LCORL* (minGwas$P \le 3.31 \times 10^{-13}$), and additional multi-population support for *DIS3L2, FAM184B,* and *CS* (S16 Table). Gene-based analysis with MAGMA pinpointed 8 significant genes ($P < 0.05/18,651$), most notably *IGF1, ATF7,* and *CCDC53* (S17 Table and S3–S5 Figs). Our FSS GWAS of 2,050 cases and 27,966 controls in Han Taiwanese individuals identified 27 genes across five loci (S13–S17 Tables). To prioritize biologically relevant candidates, we performed Ingenuity Pathway Analysis (IPA), highlighting enrichment in key growth-related processes, including Cell Cycle: G1/S Checkpoint Regulation and Growth Hormone Signaling (S18 and S19 Tables and S6 and S7 Figs).

All FSS-associated loci replicated established height loci in both East Asian and European populations. We performed sensitivity analyses using an alternative control set defined by height SDS ≥ 75th percentile and additionally matched by age and sex to address potential selection bias (S8 Fig and S12 Table). The resulting GWAS signals were highly concordant with the primary analysis (Figs 2C and S9).

The cross-trait genetic correlation analyses identified significant genetic association between FSS and multiple phenotypes across East Asian cohorts—seven in TWB, one in CMUH, none in CKB, two in BBJ, and four in KoGES—after multiple testing correction using the Bonferroni method (Fig 2D and S20-S24 Tables). Notably, a higher genetic predisposition to FSS was associated with shorter body size and diminished lung function, and showed a negative genetic association with endometriosis.

## Stature genetic clustering with health conditions

Across major East Asian biobanks, cross-trait genetic correlation analyses further identified significant associations among body height and 31 phenotypes (87 combinations), and among FSS and 14 phenotypes (44 combinations) (Fig 3A and 3B; S25 and S26 Tables). Hierarchical genetic clustering of body height revealed major clusters, including larger body size, enhanced lung function, delayed menarche, lower LDL-C levels, reduced kidney function, and a heightened AF risk. In parallel, FSS was genetically correlated with traits reflecting shorter body size and diminished lung function, and exhibited an inverse genetic correlation pattern with endometriosis, indicating a comparable correlation-based pattern with shared genetic architecture.

## Phenome-wide association study of stature PRS (PRS-PheWAS) with health conditions

For the [TWB5] HEIGHT PRS-PheWAS study (a PheWAS specifically applied to PRS), we identified an independent cohort from the CMUH_410K_SNP dataset (N = 379,783) (S10 Fig) and linked it to CMUH clinical data, yielding 374,896 individuals with both SNP and clinical records. After quality control exclusions (duplicate samples, missing phecodes, and other criteria), 296,745 participants remained for analysis. The cohort had a mean age of 46.7 years, a mean follow-up of 46.5 years, and an average height of 157.0 cm, with phenotypes defined from electronic health records (S27 Table).

We utilized GWAS summary statistics for body height loci in Han Taiwanese individuals to develop a PRS in 296,745 participants from the CMUH database (S10 Fig). Subsequently, a PRS-PheWAS was performed (i.e., applying PheWAS to PRS), adjusting for age, sex, and 10 principal components (PCs) (Fig 4A). The PheWAS of height PRS in the CMUH cohort identified 13 disease phenotypes ($P < 0.05/1,090$), with the strongest associations for lack of normal physiological development, short stature, atrial fibrillation (AF), and endometriosis (S28 Table).

For the [CMUH] FSS PRS-PheWAS study (a PRS-focused application of PheWAS), we assembled an independent, non-overlapping cohort from the CMUH_410K_SNP dataset linked to electronic health records, separate from the FSS GWAS discovery sample (S11 Fig). After excluding all individuals included in the prior FSS GWAS, removing duplicates, and filtering out participants with missing phecodes or failing quality control, 258,931 participants remained for analysis

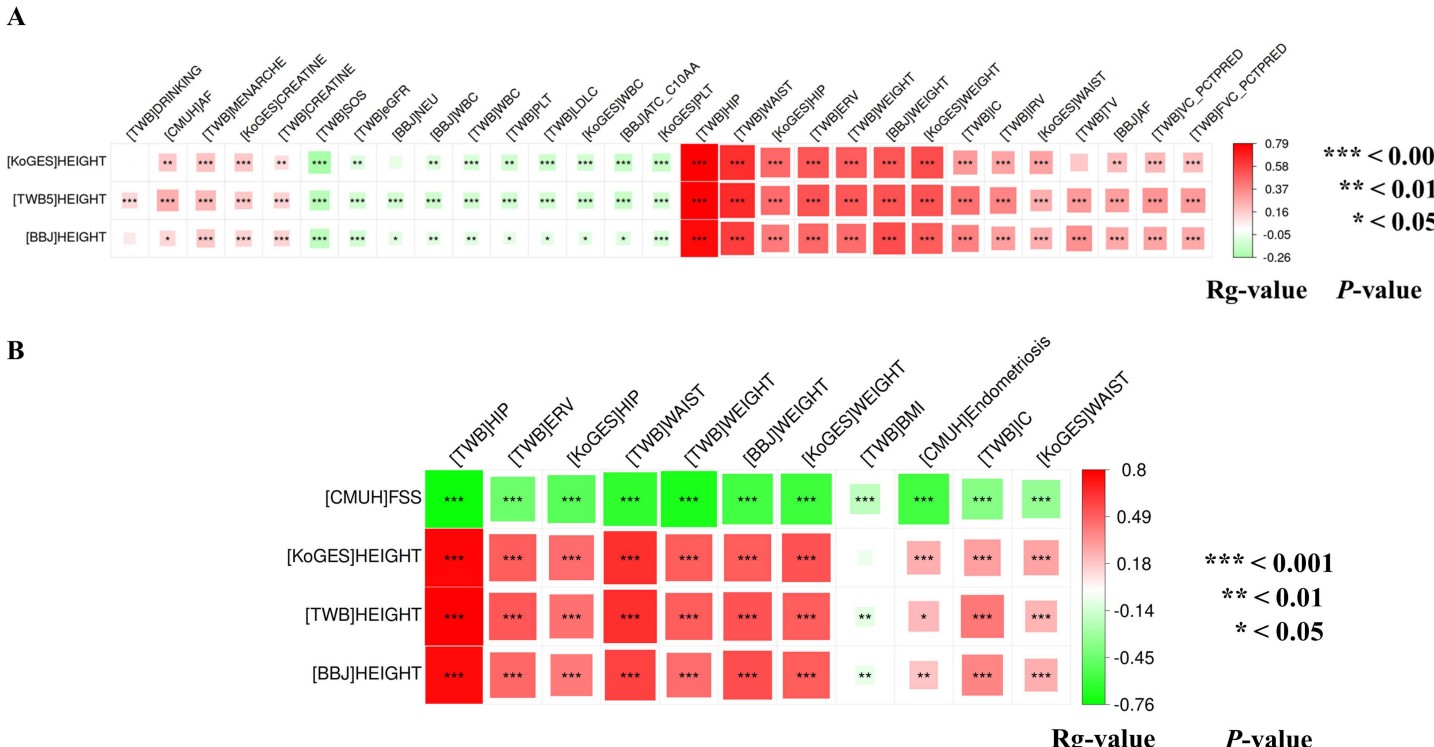

**Fig 3. Cross-trait genetic correlations between stature and associated conditions across five East Asian biobanks. A.** Hierarchical genetic clustering heatmap showing genetic correlations between [TWB5]HEIGHT and 32 related phenotypes across East Asian biobanks. Positive correlations are shown in red and negative in green, with significance levels indicated by asterisks (*P<0.05; **P<0.01; ***P<0.001). **B.** Hierarchical genetic clustering heatmap showing genetic correlations between [CMUH]FSS and 14 related phenotypes across East Asian biobanks. Positive correlations are shown in red and negative in green, with significance levels indicated by asterisks (*P<0.05; **P<0.01; ***P<0.001). Abbreviations: AF, atrial flutter/ fibrillation; ATC_C10AA, HMG-CoA reductase inhibitors (Anatomical Therapeutic Chemical code C10AA); BBJ, Biobank Japan; BMI, body mass index; CMUH, China Medical University Hospital; CKB, China Kadoorie Biobank; CREATININE, serum creatinine; DRINKING, ≥6 months of alcohol consumption; eGFR, estimated glomerular filtration rate; ERV, expiratory reserve volume; FSS, familial short stature; FVC_PCTPRED, % of predicted forced vital capacity; HEIGHT, body height; HIP, hip circumference; IC, inspiratory capacity; IRV, inspiratory reserve volume; KoGES, Korean Genome and Epidemiology Study; LDLC, low-density lipoprotein cholesterol; LDSC, Linkage Disequilibrium Score Regression; MENARCHE, age at first menstruation; NEU, neutrophil count; PLT, platelet count; SE, standard error; SOS, speed of sound (QUS); TV, tidal volume; TWB, Taiwan Biobank; VC_PCTPRED, % of predicted vital capacity; WAIST, waist circumference; WBC, white blood cell count; WEIGHT, body weight.

(S11 Fig). This cohort had a mean age of 47.1 years, an average follow-up of 46.9 years, and 1.5% of individuals carried a FSS diagnosis, with phenotypes defined using EHR-derived phecodes (S29 Table). In the CMUH cohort, PheWAS of the short-stature–specific PRS identified 3 disease phenotypes (P<0.05/1,042), with the strongest associations in abnormal growth and short stature, and nominal associations in endometriosis (Fig 4B and S30 Table).

To replicate our CMUH findings in independent populations, we constructed PRSs from stature and GWAS meta-analysis data and validated them in CMUH and BBJ. PRSs derived from height-related GWAS meta-analyses showed consistent associations across cohorts (S31 Table). Higher height PRSs were significantly associated with increased atrial fibrillation (AF) risk in both Biobank Japan (BBJ; OR = 1.14–1.15, $P<2.0\times10^{-16}$) and CMUH (OR = 1.12, $P=3.0\times10^{-18}$). A positive association was also observed for endometriosis in CMUH (OR = 1.07, $P=4.8\times10^{-9}$), though not in BBJ. FSS-derived PRS showed a non-significant trend for endometriosis in BBJ.

To test whether the height PRS adds predictive value beyond disease-specific PRSs, we performed conditional logistic regression to assess its independent contribution to AF and endometriosis risk (S32 Table). In the CMUH cohort, the

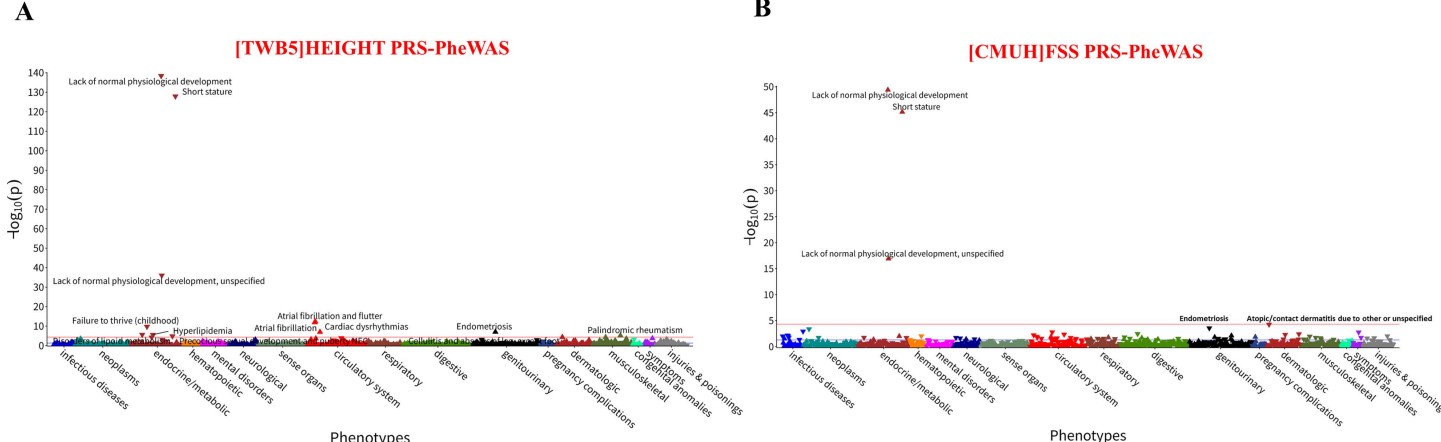

**Fig 4. Phenome-wide association of stature polygenic risk scores with related health conditions. A.** PheWAS Manhattan plot for the body height PRS across 1,090 disease phenotypes in 295,829 participants. The horizontal line denotes the Bonferroni-corrected threshold ($P < 0.05/1,090$). Significant associations are labeled. Upward triangles indicate positive (OR > 1) and downward triangles inverse (OR < 1) associations. **B.** PheWAS Manhattan plot for the FSS PRS across the same 1,042 disease phenotypes, with the Bonferroni threshold indicated. Upward triangles indicate positive (OR > 1) and downward triangles inverse (OR < 1) associations. Abbreviations: CMUH, China Medical University Hospital; FSS, familial short stature; PheWAS, phenome-wide association study; OR; odds ratio; PRS, polygenic risk score..

height PRS remained significantly associated with AF after adjustment for the AF PRS (**Model 2**: OR = 1.09 per SD-PRS, $P = 1.43 \times 10^{-11}$), indicating predictive value beyond the disease-specific PRS. In contrast, its modest association with endometriosis was attenuated after adjustment for the endometriosis PRS (**Model 2**: OR = 1.01 per SD-PRS, $P = 0.45$), suggesting no independent predictive contribution.

## Causal effect of stature on endometriosis

To assess whether stature contributes independently to endometriosis risk, we performed conditional analyses showing that the height PRS association was attenuated after adjustment for the endometriosis PRS (S32 Table). Given this attenuation and the paradoxical links among stature, menarche, obesity, and endometriosis [27–30], we applied MR analyses to test whether the effect of stature on endometriosis is direct or mediated through menarche timing or body weight, thereby clarifying the genetic contribution to this discrepancy.

Univariable MR showed no evidence for a causal effect of height on endometriosis ($P > 0.05/5$) (S12A Fig, left, and S33–S35 Tables). In contrast, following multivariable adjustment, significant associations emerged for age at menarche (MVMR-Lasso: OR = 1.112, $P = 9.07 \times 10^{-3}$) and body weight (OR = 1.165, $P = 4.63 \times 10^{-3}$), indicating stronger contributions of these traits to endometriosis risk than height (S12B–S12C Fig, left, and S33 and S34 Tables). Likewise, while FSS showed no effect in univariable MR or after adjustment for age at menarche, it was significant after accounting for body weight (MVMR-Robust: OR = 0.903, $P = 6.68 \times 10^{-3}$) (S12A–S12C Fig, right) (S33–S35 Tables).

## Causal effect of body height on atrial flutter/fibrillation

Univariable MR demonstrated a significant causal effect of height on atrial fibrillation (MR-weighted median: OR = 1.32, $P = 9.68 \times 10^{-5}$) (S13A Fig and S36–S38 Tables). This association persisted after adjustment for age at menarche (MVMR-Lasso: OR = 1.31, $P = 6.51 \times 10^{-9}$) and body weight (OR = 1.24, $P = 4.07 \times 10^{-4}$), indicating that body height exerts an independent contribution to AF risk (S13B–S13C Fig and S37 and S38 Tables).

## Applications of stature in genetics

Given the strong PheWAS and MR associations between height PRS and atrial flutter/fibrillation, but not endometriosis (Figs 4, S12, and S13 and S33–S38 Tables), we next assessed the longitudinal predictive value of height PRS for atrial flutter/fibrillation. Cox proportional hazards models and Kaplan–Meier survival analyses were employed to evaluate these associations and the predictive power of body height PRS (Fig 5). The left panel showed Cox proportional hazards models indicating that individuals in the highest PRS quintile had a significantly increased AF risk compared with the middle PRS quintile (adjusted hazards ratio [HR]: 1.09, $P = 1.72 \times 10^{-02}$). Similarly, the top 20% ($PRS_{80-100\%}$) had a higher AF risk than the remaining population ($PRS_{0-80\%}$) (adjusted HR = 1.16, 95% CI: 1.10–1.23, $P = 2.70 \times 10^{-7}$). The adjusted HR per SD-PRS was 1.10 (95% CI = 1.07-1.13, $P = 9.80 \times 10^{-15}$).

The right panel showed the divergence of the cumulative incidence curves at the 10% incidence threshold, with high-risk individuals ($PRS_{80-100\%}$) exhibiting an earlier divergence point than low-risk individuals ($PRS_{0-20\%}$), evident in the ages at which this threshold was reached (78.2 vs. 81.6 years). This divergence was statistically significant across groups, as confirmed by log-rank tests (High vs Low: $P = 5.48E-11$; High vs Middle: $P = 3.05E-02$; Middle vs Low: $P = 1.11E-05$), with high-PRS individuals showing a steeper cumulative-incidence trajectory and an earlier divergence of the curves. These cumulative-incidence patterns were consistently replicated in survival analyses using meta-analysis–derived PRSs (S14–S16 Figs and S39–S41 Tables). Overall, these findings underscore the clinical relevance of stature-related genetic risk in predicting key disease outcomes.

## Discussion

We identified 293 loci for body height and 5 for FSS in Han Taiwanese, including 16 novel height loci, highlighting shared genetic pathways such as cell cycle G1/S checkpoint regulation and growth hormone signaling. The results of genetic

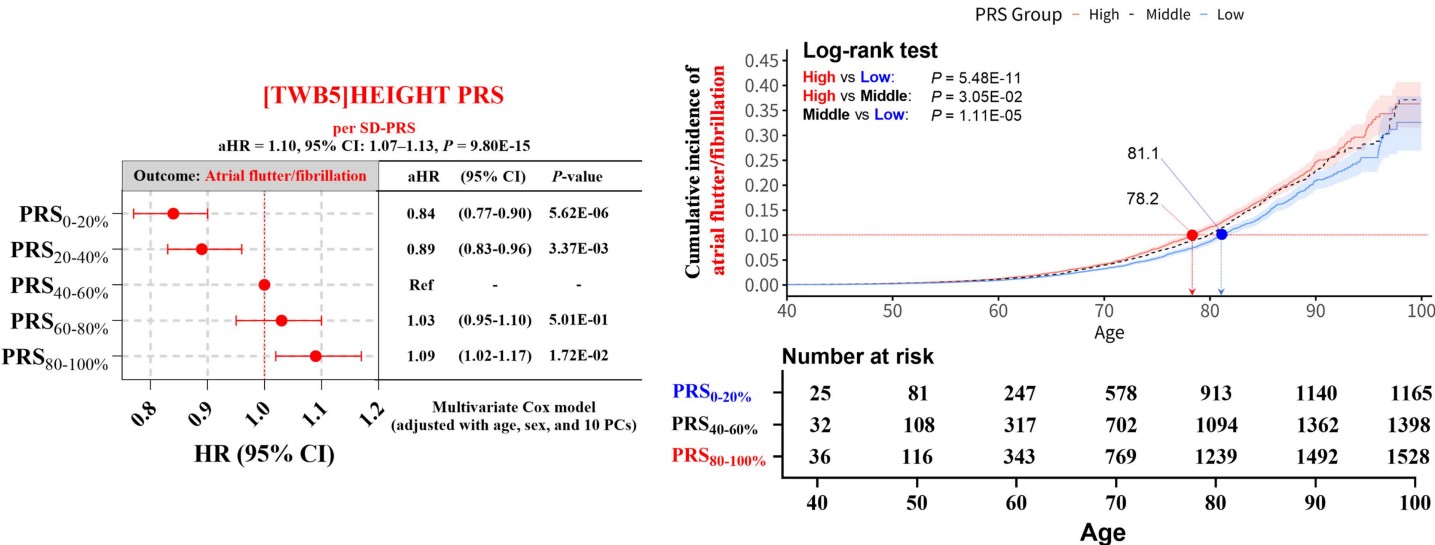

**Fig 5. Cox proportional hazards model and cumulative incidence of atrial flutter/fibrillation by stature PRS quintiles. Left panel:** Multivariate Cox model results for atrial flutter/fibrillation risk across body height PRS quintiles, presenting hazard ratios (HRs) and 95% confidence intervals (CIs) relative to the reference group ($PRS_{40-60\%}$), adjusted for age, sex, and 10 principal components. **Right panel:** Kaplan–Meier survival curves illustrating the cumulative incidence of atrial flutter/fibrillation among individuals with low ($PRS_{0-20\%}$), middle ($PRS_{40-60\%}$), and high ($PRS_{80-100\%}$) genetic risk. P-values are based on log-rank tests comparing differences across PRS groups. The dashed line denotes the age at which each PRS group reaches a cumulative incidence of 10%. Numbers displayed below the curves represent participants at risk at each time point for the corresponding groups. Abbreviations: AF, atrial flutter/fibrillation; CI, confidence interval; HR, hazard ratio; PCs, principal components; PRS, polygenic risk score.

correlation analyses associated height and FSS with 14–32 traits, primarily related to body size, lung function, and cardiovascular/reproductive health, including AF, menarche, and endometriosis. PheWAS analysis revealed that tall stature increased risks of AF and endometriosis, whereas short stature had a weak protective effect against endometriosis. MR analyses showed that taller stature increased AF risk independently and endometriosis risk through menarche/weight, while shorter stature had a weak protective effect against endometriosis. Importantly, PRS-based time-to-event analyses showed that individuals with higher height PRS exhibited not only a greater overall risk of AF but also an earlier divergence of cumulative incidence curves, reflecting differences in time-to-event patterns rather than age at onset. These cumulative-incidence patterns were consistently replicated in survival analyses using meta-analysis–derived PRSs, supporting the robustness of the observed associations. These findings suggest genetic links between stature and selected health outcomes in East Asians, supporting the use of stature-related PRS in early disease risk prediction and precision health.

We conducted a GWAS of height in 120,301 Han Taiwanese individuals, identifying 293 loci and 1,305 genes (1,185 from GWAS annotations, 619 from gene-based GWAS) (S2-S6 Tables and Figs 2A and S2). A meta-analysis of our Han Chinese and Korean cohorts revealed 433 loci and 1,690 genes, including 1,563 from GWAS annotations and 967 from gene-based analyses (S39-S41 Tables) (S14 and S15 Figs). In the CMUH GWAS of 2,050 FSS cases and 27,966 controls, we detected five loci and 27 genes (26 from GWAS annotations, 8 from gene-based GWAS), highlighting their importance in FSS and stature genetics (S18 and S19 Tables and Figs 2C and S5). Pathway analyses highlighted key processes including cell cycle G1/S checkpoint regulation and growth hormone signaling (S6 and S7 Figs). Using a stringent three-stage verification framework integrating conditional association analysis and linkage disequilibrium (LD) profiling, we identified 16 novel loci (S3 Table), including *GIGYF2*, *MAP3K13*, *NSD2*, *THBS2*, *EXTL3*, *ZFAT*, *FBP1*, *LTBP3*, *SPRY2*, *SLC12A6*, *PIEZO1*, *LNCNEF*, *SNX21*, and three uncharacterized regions. Functional annotation underscores the biological plausibility of these associations. For instance, *THBS2* and *PIEZO1* regulate skeletal matrix integrity [31] and mechanotransduction [32], respectively, while *EXTL3* and *FBP1* are critical for heparan sulfate biosynthesis [33] and chondrogenic metabolism [34]. Other identified loci modulate key developmental signaling pathways: *MAP3K13* regulates MAPK signaling [35]; *SPRY2* fine-tunes RTK and BMP signaling in endochondral bone [36], and *NSD2* has been implicated in developmental delays [37–39]. Our findings revealed height- and FSS-associated loci as potential essential mechanisms governing skeletal homeostasis and growth.

Our genetic correlation analyses across major East Asian biobanks identified significant associations between height with 32 traits and FSS with 14 traits, clustering into body size, lung function, and cardiovascular/reproductive traits (AF, menarche, and endometriosis), highlighting shared genetic influences on growth and health-related conditions. Body height is genetically linked to various conditions, including cardiovascular disease, cancer, and metabolic disorders [21,40]. Specifically, Raghavan *et al.* reported in their Mendelian randomization-PheWAS analysis that genetically predicted body height is linked to an increased risk of AF but a decreased risk of coronary heart disease, hypertension, and hyperlipidemia—conditions that critically affect older adults [21]. Vithayathil *et al.* reported in their Mendelian randomization analysis that genetically predicted height increases the risk of overall and site-specific cancers, including kidney, colorectal, biliary tract, breast, and ovarian cancers [40]. Our findings further reveal significant genetic correlations between body height, FSS, and various health conditions across East Asian biobanks, highlighting complex genetic interactions.

In contrast to our previous GWAS of height in CMUH [9], which used the CMUH_250K_SNP dataset (2018–2021) and PRSice-2 for PRS construction, the current study leveraged the expanded CMUH_410K_SNP dataset (2018–2023) (S3 Fig) and applied PRS-CS with the 1000 Genomes Phase 3 East Asian LD reference panel [41]. The present analysis also included 120,301 TWB participants compared with 67,452 in the earlier study [9], thereby improving statistical power. Whereas the prior work emphasized associations of genetically determined height with anthropometric and metabolic traits [9], our study extended these findings through genetic correlation, PheWAS, and PRS analyses, consistently linking stature-related genetic variants with risks of AF and endometriosis. Replication analyses in CMUH and BBJ confirmed

the associations, with height and meta-analysis–derived PRSs positively associated with atrial fibrillation in both cohorts and with endometriosis in CMUH only, while FSS PRSs showed no significant association with endometriosis in BBJ (S31 Table). Survival analyses showed that individuals in the highest height-PRS strata had significantly elevated AF risk and steeper cumulative-incidence trajectories compared with lower-PRS groups (height: HR per SD = 1.10, 95% CI: 1.07–1.13, $P = 9.80 \times 10^{-15}$; meta-analysis: HR = 1.11, 95% CI: 1.09–1.14, $P = 1.16 \times 10^{-18}$) (Figs 5 and S16), highlighting the clinical relevance of stature genetics. Mechanistically, increased body height correlates with larger left atrial size, a known independent risk factor for AF [42], along with the structural and functional parameters of the left atrium, including volume index, expansion index, and contractile function [43]. Regarding endometriosis, genetically influenced traits such as central obesity may increase the risk by increasing leptin levels from adipocytes, promoting inflammation and hormonal dysregulation contributing to disease development [44–46].

Epidemiologic studies have reported complex and sometimes inconsistent associations between stature, menarche, and endometriosis [27–30]. While taller stature and earlier menarche have been linked to an increased risk of endometriosis [27,29], other studies suggest that childhood obesity may play a protective role [30], despite its known correlation with earlier menarche [28]. In our MR analyses, we found no direct causal effect of height or FSS on endometriosis risk; however, associations emerged after adjusting for menarche or body weight, suggesting that these traits exert stronger influences on risk than stature itself. These findings highlight suppression and pleiotropy across shared genetic pathways and suggest that the apparent paradox reflects mediation through growth, reproductive, and metabolic factors rather than direct effects of body height or FSS.

Our study has some limitations. First, because the findings are based on Han Taiwanese and other East Asian populations, they are population-specific. However, they may serve to replicate associations previously reported in European cohorts. Second, while we identified novel loci, further functional studies are necessary to validate their roles in growth biology and long-term health outcomes.

In conclusion, the analysis of large-scale GWAS of Han Taiwanese individuals resulted in the identification of 293 loci associated with adult height and 5 with FSS, highlighting the complex genetic architecture of stature and its associations with body size, lung function, and cardiovascular and reproductive traits. Herein, we integrated GWAS, PheWAS, and PRS-based analyses, indicating that genetically taller stature is associated with an increased risk of AF, whereas no significant association was observed with endometriosis after correction for multiple testing. Altogether, these findings support the potential utility of stature-based polygenic scores for disease risk stratification and precision health in East Asian populations. Nevertheless, further research is needed to validate these associations and inform clinical applications.

## Materials and methods

### Ethics statement

Ethical approval was obtained from the Research Ethics Committee of China Medical University Hospital (CMUH107-REC3–074[CR-7] and CMUH111-REC1–176[CR-3]).

### Study participants

For the body height GWAS, data from the TWB (https://www.twbiobank.org.tw/new_web/) [9] was used, comprising demographic, lifestyle, and physical measurements, along with SNP genotyping and biosample data. All information was anonymized; therefore, additional informed consent was not required.

From 189,132 Taiwan Biobank participants, we excluded 42,757 without GWAS data, 25,985 failing QC or PCA, 62 with missing height, and 27 with extreme values (>±4 SD) (S1 Fig). After these exclusions, 120,301 participants remained for the final GWAS. Quality control excluded individuals with >2% missing genotypes, heterozygosity beyond ±5 SD, kinship >0.0884, or divergent ancestry with Euclidean distance beyond ±5 SD, and SNPs with >2% missingness, Hardy-Weinberg equilibrium (HWE) $P < 1 \times 10^{-5}$, or minor allele frequency (MAF) <0.001. Height (cm) was analyzed as a quantitative trait.

Measurements were stratified by sex, Z-scores normalized within each sex group, and subsequently recombined for downstream analyses. We performed the GWAS using REGENIE (https://github.com/rgcgithub/regenie) [47] under a mixed linear model, adjusting for age, sex, and 10 PCs. Genome-wide significance was defined as $P < 5 \times 10^{-8}$.

S3 Fig shows the CMUH SNP database timeline, with 247,249 participants recruited in the CMUH_250K_SNP dataset (2018–2021) and 379,783 in the CMUH_410K_SNP dataset (2018–2023); the recruitment is ongoing. FSS cases and controls were identified from CMUH electronic health records containing both SNP and clinical data (S4 Fig). Familial short stature (FSS) is intrinsically a pediatric condition, and all cases were of children diagnosed by pediatric endocrinologists in Taiwan under stringent criteria, including height <3rd percentile, at least one parent with height <3rd percentile, bone age concordant with chronological age, normal pubertal onset, normal annual growth rate, and unremarkable clinical biochemistry. Controls were selected from the CMUH Biobank as a definitively unaffected comparison group rather than age-matched children. We restricted controls to adults (>18 years) with finalized stature ≥75th percentile for age and sex in Taiwan and no history of FSS to avoid misclassification from ongoing growth, thereby maximizing phenotype certainty and contrast.

For FSS GWAS, the data from the CMUH Biobank (http://cmuh-biobank.eastasia.cloudapp.azure.com:6001/#/gwas_list) was analyzed, which contained genetic, laboratory, and demographic records from 1996 to 2021. SNPs were genotyped using the customized Axiom TPM v1 array (~714,000 variants, equivalent to TWB v2) [48], with stringent quality control at both variant and individual levels, followed by phasing and imputation with BEAGLE using TWB WGS as the reference panel [49,50]. Post-imputation, variants with an INFO score ≥ 0.3 were retained. Quality control for array data was conducted using PLINK (version 2.0) [51].

Prior to downstream GWAS, we re-applied quality control to the imputed SNP data. SNP quality control excluded variants with a minor allele frequency < 0.001, a Hardy-Weinberg equilibrium $P$-value $< 1 \times 10^{-6}$, or a missing call rate > 5%. The remaining SNPs were used for PC analysis to assess population structure. Individual quality control excluded participants who failed the sex check (male < 0.75, female < 0.25), had a heterozygosity rate exceeding ±3 standard deviations (SD), a missing call rate > 5%, or divergent ancestry based on Euclidean distance (outliers beyond ±5 × SD). Individuals with kinship coefficients > 0.0884 were also removed. Diagnoses were coded using the International Classification of Diseases, 9th Revision, Clinical Modification and PheCODE [52], with follow-up from birth until the first PheCODE-defined disease or last data entry (S28 and S30 Tables). Cases were defined as ≥2 outpatient visits or one inpatient admission, and diseases with fewer than 200 cases were excluded. Of 379,783 participants, we excluded those with missing data or failing quality control, retaining 2,050 FSS cases and 27,966 controls. The GWAS used REGENIE [47] with a case-control model, adjusting for age, sex, and the first 10 PCs, and adopted $P < 5 \times 10^{-8}$ as the genome-wide significance threshold.

## Locus definition

Distinct loci were defined using LocusZoom (locuszoom.org) with default settings. For each genome-wide significant lead SNP, we defined the locus as all variants within ±500 kb of the lead SNP and in linkage disequilibrium (LD) at r² ≥ 0.4, based on the 1000 Genomes Phase 3 East Asian reference panel. Loci whose intervals overlapped were merged, and only non-overlapping regions were counted as distinct loci.

## GWAS summary statistics

We performed GWAS analyses for body height using summary statistics from the TWB and FSS using the CMUH database, following established protocols [9,50,53,54]. To avoid sample overlap and facilitate cross-population comparisons, we further leveraged GWAS summary statistics from five East Asian biobanks—BBJ (https://pheweb.jp/) [20], KoGES (https://koges.leelabsg.org/) [26], TWB (https://pheweb.twbiobank.org.tw:5038/about), CMUH (http://cmuh-biobank.eastasia.cloudapp.azure.com/), and CKB (https://pheweb.ckbiobank.org/)—ensuring non-overlapping East Asian populations for unbiased cross-cohort analyses.

These GWAS summary statistics were used for downstream analyses, including conditional analysis, co-localization, and meta-analysis to validate findings (S3, S4, S14, and S15 Tables) (S14-S16 Figs), and cross-trait genetic correlations between stature and diverse diseases using linkage disequilibrium (LD) score regression across the five East Asian biobanks (Figs 1 and 2).

## Classification of novel and known loci

We implemented a stringent three-stage verification framework integrating conditional association analysis and linkage disequilibrium (LD) profiling to systematically classify the identified loci. First, we compiled a comprehensive reference catalog of published height-associated index SNPs from the GWAS Catalog and major multi-ancestry meta-analyses, including [EUR]HEIGHT, [BBJ]HEIGHT, [KoGES]HEIGHT, [Yengo_EUR]HEIGHT, and [Yengo_EAS]HEIGHT [5,20,26].

We refined the TWB5 signals as follows: (1) Internal independence: we performed GCTA-COJO joint analysis (https://github.com/kheilbron/cojo_pipe) [55] to isolate independent lead SNPs within the TWB5 dataset; (2) External novelty: we conducted conditional analyses (GCTA-COJO --cond) adjusting for the compiled reference index SNPs to assess statistical independence, prioritizing signals that retained genome-wide significance ($P_{cond} < 5 \times 10^{-8}$); and (3) Physical verification: we quantified pairwise LD ($r^2$) using PLINK 2.0 within a broad ±10 Mb window around each lead SNP to exclude long-range LD masking.

Loci were strictly stratified based on these metrics: variants demonstrating significant conditional independence ($P_{cond} < 5 \times 10^{-8}$) were defined as novel loci if they exhibited negligible LD ($r^2 < 0.1$) with known signals, or as independent secondary signals if they showed intermediate LD ($0.1 \leq r^2 < 0.6$). Conversely, variants that were statistically explained by the model ($P_{cond} \geq 5 \times 10^{-8}$) or displayed high LD ($r^2 \geq 0.6$) with previously reported associations were classified as known signals (S3 and S14 Tables).

## SNP-based gene annotation using FUMA

SNP-based gene annotation was performed using the SNP2GENE module of FUMA (https://fuma.ctglab.nl/snp2gene), which processes GWAS summary statistics to identify and functionally annotate trait-associated SNPs. The workflow included: (1) genomic loci characterization to identify independent significant SNPs using default settings; (2) functional annotation of candidate SNPs using ANNOVAR (ANNOtate VARiation), CADD, RegulomeDB, chromatin states (127 tissue/cell types), eQTLs, Hi-C chromatin interactions, and the GWAS Catalog; and (3) functional gene mapping via positional mapping (based on proximity), eQTL mapping (based on gene expression), and chromatin interaction mapping (based on physical contacts) (https://fuma.ctglab.nl/tutorial). All analyses used the 1000 Genomes Phase 3 East Asian reference panel. Results were compiled in a SNP-based gene annotation table with metrics such as posMapSNPs and minGwasP (S5, S16, and S40 Tables).

## Gene-based GWAS analysis using MAGMA implemented in FUMA

Gene-based GWAS was performed using MAGMA within the SNP2GENE module of FUMA (https://fuma.ctglab.nl/snp-2gene), which aggregates SNP-level associations into gene-level signals. SNPs were mapped to protein-coding genes using Ensembl identifiers with default windows and analyzed in MAGMA v1.06 under the SNP-wise mean model; later FUMA updates (v1.07 and v1.08) modified output format but not analytical principles. Significant susceptibility genes were identified at a Bonferroni threshold of $P < 0.05/18,651$, and results were summarized in a gene-based GWAS table including NSNPS, NPARAM, N, ZSTAT, and $P$-values (S6, S17, and S41 Tables).

## Linkage disequilibrium score regression (LDSC) analysis

We applied LDSC to estimate SNP-based heritability (h²_SNP) and cross-trait genetic correlation (Rg) for binary GWAS summary statistics [56]. Herein, precomputed linkage disequilibrium scores from 1,495 TWB whole-genome sequences

representing an East Asian reference panel (https://taiwanview.twbiobank.org.tw/browse38) were used. LDSC was performed with default settings (https://github.com/bulik/ldsc) [56], employing weighted regression of Z-statistics. Rg, ranging from −1–1, indicated negative or positive correlations.

## Instrument selection and Mendelian randomization

From GWAS summary data, we defined genetic instruments as variants reaching genome-wide significance ($P < 5 \times 10^{-8}$) and retained only near-independent signals after East Asian–specific LD clumping ($r^2 < 0.001$ within 10 Mb). Variants with MAF ≤ 0.01 were removed. Instrument strength was checked using F statistics (F > 10), and residual heterogeneity across instruments was quantified with Cochran's Q test [57]. We evaluated horizontal pleiotropy using the MR-Egger intercept and identified influential outliers with MR-PRESSO (https://github.com/rondolab/MR-PRESSO) [58].

For univariable two-sample MR, IVW served as the primary estimator, with robustness assessed via MR-Egger, weighted median, simple and weighted mode, and MR-PRESSO outlier-corrected models implemented in TwoSampleMR (https://mrcieu.github.io/TwoSampleMR/) [59]. Multivariable MR sensitivity analyses were performed using IVW and Egger extensions alongside robust, median-based, and lasso approaches, following established MVMR workflows (https://wspiller.github.io/MVMR/articles/MVMR.html#step-4-test-for-horizontal-pleiotropy-using-conventional-q-statistic-estimation-1 and https://github.com/aj-grant/robust-mvmr) [57].

## PRS calculation

PRS was generated using the PRS-CS pipeline [41] with the 1000 Genomes Phase 3 East Asian LD reference panel. Posterior SNP effect sizes were estimated from GWAS summary statistics ([TWB5]HEIGHT, [CMUH]FSS, or [TWB5 + KoGES]HEIGHT), the external LD reference panel, and target cohort genotypes (CMUH or BBJ), yielding ~401K SNPs for HEIGHT, ~642K for FSS, and ~386K for the GWAS-meta HEIGHT PRS calculation [51]. PRSs were computed in CMUH or BBJ cohorts using PLINK v2.0 with East Asian–specific weights and normalized within each validation cohort to avoid bias [51]. We applied a fixed global shrinkage parameter ($\phi = 1 \times 10^{-2}$), as recommended for polygenic traits like height and suitable for our moderate sample size, to improve predictive accuracy. Default gamma–gamma prior settings (a = 1.0, b = 0.5) and Markov chain Monte Carlo (MCMC) configuration (1,000 iterations, 500 burn-in, and a thinning factor of 5) were used, given their computational efficiency and appropriateness for polygenic modeling.

## PheWAS

Although PheWAS can also refer to pleiotropic scans of specific variants or broad phenome-wide GWAS, in this study we specifically performed PRS-PheWAS analyses, applying PheWAS methods to polygenic risk scores. Here, PheWAS was applied to polygenic risk scores rather than single variants or genome-wide scans. A PheWAS was performed to assess associations between Z-score–standardized stature PRS and disease phenotypes in the CMUH cohort. Multivariate logistic regression was applied, adjusting for age, sex, and the first 10 PCs. Diagnoses were coded using the International Classification of Diseases, 9th Revision, Clinical Modification and mapped to PheCODEs [52]; cases were defined by ≥2 outpatient visits or one inpatient admission. Diseases with <200 cases were excluded. Statistical significance was determined using a Bonferroni correction.

## Statistical analyses

GWAS analysis was performed with REGENIE (https://github.com/rgcgithub/regenie), adjusting for age, sex, and the first 10 PCs [47]. We conducted colocalization analysis using the coloc R package (https://cran.r-project.org/web/packages/coloc/vignettes/a01_intro.html)[25]. To assess the clinical impact of genetically predicted stature, we conducted survival analyses of stature PRS quintiles (Q1–Q5) and atrial fibrillation (AF; PheCODE 427.2) using Cox regression to estimate HRs and 95% confidence intervals (CIs), with time-to-event defined as the interval from cohort entry to first AF diagnosis

or censoring at last follow-up. Participants without AF during the observation period were censored, and models were adjusted for age, sex, and PCs. Kaplan–Meier curves compared AF risk across PRS quintiles, emphasizing the highest (Q5) versus lowest (Q1), with log-rank tests and 95% confidence bands used to evaluate group differences. All statistical analyses were performed using Python, R, PLINK (version 2.0), REGENIE, and SAS (version 9.4; SAS Institute, Cary, NC, USA).

## Supporting information

**S1 Fig. Flowchart illustrating the selection process for 120,301 participants included in the genome-wide association study (GWAS) of body height in the Taiwan Biobank.** Starting from 189,132 TWB participants, individuals were excluded based on the following criteria: missing GWAS data (N = 42,757), failure to pass GWAS quality control (QC) and principal component analysis (PCA) (N = 25,985), missing height information (N = 62), and extreme height values beyond ±4 standard deviations (SD) from the mean (N = 27).
(TIF)

**S2 Fig. Manhattan and QQ plots for the gene-based GWAS of body height of 120,301 Han Taiwanese in TWB (corrected threshold: P = 0.05/18,306).**
(TIF)

**S3 Fig. Timeline of participant recruitment and genome-wide SNP genotyping for the CMUH SNP database.** The first genotyping phase (August 2018–December 2021) included 247,249 participants, and the second phase (August 2018–March 2023) included 379,783 participants. The CMUH_250K_SNP dataset corresponded to data generated during the first phase, while the CMUH_410K_SNP dataset corresponded to the expanded data collection. Abbreviations: CMUH, China Medical University Hospital; SNP, single-nucleotide polymorphism.
(TIF)

**S4 Fig. Flowchart of study subject selection for the FSS GWAS in the CMUH database.** The CMUH_250K_ SNP dataset comprised 247,249 participants with genotyping data, and the CMUH_clinical dataset included 297,915 participants with electronic health records. A total of 142,935 participants had both SNP and clinical data, from which 2,050 familial short stature (FSS) cases and 27,966 controls were identified for genome-wide association analysis.
(TIF)

**S5 Fig. Manhattan and QQ plots of the gene-based GWAS for 2,050 FSS cases and 27,966 controls in the CMUH cohort (significance threshold: P = 0.05/18,306).**
(TIF)

**S6 Fig. Ingenuity Pathway Analysis (IPA) of the canonical pathway Cell Cycle: G1/S Checkpoint Regulation associated with genes from [TWB5]HEIGHT, [TWB5+KoGES]HEIGHT, and [CMUH]FSS.** Abbreviations: IPA, Ingenuity Pathway Analysis; TWB5, Taiwan Biobank (version 5); Korean Genome and Epidemiology Study; CMUH, China Medical University Hospital; FSS, familial short stature.
(TIF)

**S7 Fig. Ingenuity Pathway Analysis (IPA) of the canonical pathway Growth Hormone Signaling associated with genes from [TWB5]HEIGHT, [TWB5+KoGES]HEIGHT, and [CMUH]FSS.** Abbreviations: IPA, Ingenuity Pathway Analysis; TWB5, Taiwan Biobank (version 5); Korean Genome and Epidemiology Study; CMUH, China Medical University Hospital; FSS, familial short stature.
(TIF)

**S8 Fig. Flowchart of subject selection for the FSS GWAS in the CMUH database.** Data integration of the CMUH_250K_SNP (N = 247,249) and CMUH_clinical (N = 297,915) datasets identified 142,935 subjects with overlapping records. Following the exclusion of individuals with prior study involvement or specific diagnoses (e.g., skeletal dysplasia, dysmorphic features, and abnormal thyroid/puberty function), 142,004 subjects remained. To generate a robust control set, individuals aged > 18 years or with height SDS < 75th percentile were removed. Constrained nearest neighbor matching (1:1 ratio, age ± 1 year, sex-matched) was performed to balance the final dataset, resulting in 2,028 FSS cases and 2,028 matched controls.
(TIF)

**S9 Fig. Genome-wide association study (GWAS) of familial short stature (FSS) using re-defined controls.** Manhattan and Quantile-Quantile (QQ) plots summarized the sensitivity analysis comparing 2,028 FSS cases with 2,028 controls from the CMUH cohort. Controls were selected via constrained nearest neighbor matching (1:1 ratio, age ± 1 year, sex-matched) to balance the dataset. The red dashed line indicated the genome-wide significance threshold ($P < 5 \times 10^{-8}$), with the *NCAPG* locus labeled. The inset displayed the QQ plot of observed versus expected $-\log_{10}(P)$ values.
(TIF)

**S10 Fig. Flowchart of participant selection for the independent CMUH cohort used in the [TWB5] HEIGHT PRS-PheWAS study.** Starting with 379,783 individuals from the CMUH_410K_SNP dataset and 437,346 from the CMUH clinical database, 374,896 participants with both SNP and clinical data were identified. After exclusions for duplicate genetic samples, missing 1,866 phecodes in disease definitions, or other QC-based removals (N = 78,151), a total of 296,745 individuals were retained for the final analysis cohort. Abbreviations: CMUH, China Medical University Hospital; TWB5, Taiwan Biobank (version 5); PRS, polygenic risk score; PheWAS, phenome-wide association study; SNP, single-nucleotide polymorphism; QC, quality control.
(TIF)

**S11 Fig. Flowchart of participant selection for the independent CMUH cohort used in the FSS PRS-PheWAS study.** From 379,783 individuals in the CMUH_410K_SNP dataset and 407,753 in the CMUH clinical database, 360,677 with both SNP and clinical data were identified. After exclusions for prior FSS GWAS participation, duplicate samples, missing 1,866 phecodes, and other QC-based removals, 258,931 participants remained for analysis. Abbreviations: CMUH, China Medical University Hospital; FSS, familial short stature; PRS, polygenic risk score; PheWAS, phenome-wide association study; SNP, single-nucleotide polymorphism; GWAS, genome-wide association study; QC, quality control.
(TIF)

**S12 Fig. Causal effect of stature on endometriosis in East Asians assessed by two-sample and multivariate Mendelian randomization (adjusted for body weight or age at menarche).** A. Causal effect of stature on endometriosis in East Asians using two-sample Mendelian randomization (MR). Left: Effect of [TWB5] HEIGHT on [CMUH] Endometriosis; Right: Effect of [CMUH] familial short stature (FSS) on [CMUH] endometriosis. B. Causal effect of stature on endometriosis in East Asians using multivariate MR adjusted for age at menarche. Left: Effect of [TWB5] HEIGHT on [CMUH] Endometriosis; Right: Effect of [CMUH] familial short stature (FSS) on [CMUH] endometriosis. [TWB5] MENARCHE was included as the covariate in MVMR. C. Causal effect of stature on endometriosis in East Asians using multivariate MR adjusted for body weight. Left: Effect of [TWB5] HEIGHT on [CMUH] Endometriosis; Right: Effect of [CMUH] familial short stature (FSS) on [CMUH] endometriosis. [TWB5] WEIGHT was included as the covariate in MVMR. Abbreviations: TWB5, Taiwan Biobank (version 5); HEIGHT, body height; CMUH, China Medical University Hospital; FSS, familial short stature; MR, Mendelian randomization; MVMR, multivariate MR; WEIGHT, body weight; MENARCHE, age at menarche.
(TIF)

**S13 Fig. Causal effect of body height on atrial flutter/fibrillation in East Asians assessed by two-sample and multivariate Mendelian randomization (adjusted for body weight or age at menarche).** A. Causal effect of body height on atrial flutter/fibrillation in East Asians using two-sample Mendelian randomization (MR). Effect of [TWB5] HEIGHT on [BBJ]AF. B. Causal effect of body height on atrial flutter/fibrillation in East Asians using multivariate MR adjusted for age at menarche. Effect of [TWB5] HEIGHT on [BBJ]AF. [TWB5] MENARCHE was included as the covariate in MVMR. C. Causal effect of body height on atrial flutter/fibrillation in East Asians using multivariate MR adjusted for body weight. Effect of [TWB5] HEIGHT on [BBJ]AF. [TWB5] WEIGHT was included as the covariate in MVMR. Abbreviations: TWB5, Taiwan Biobank (version 5); BBJ, Biobank Japan; HEIGHT, body height; AF, atrial flutter/fibrillation; MR, Mendelian randomization; MVMR, multivariate MR; WEIGHT, body weight; MENARCHE, age at menarche. (TIF)

**S14 Fig. Manhattan and QQ plots for the body height GWAS of Han Taiwanese and Korean cohorts, showing −$\log_{10}$(P) across chromosomes (P<5 x 10-8).**
(TIF)

**S15 Fig. Manhattan and QQ plots the gene-based GWAS of body height of Han Taiwanese and Korean cohorts, showing −$\log_{10}$(P) across chromosomes (significance threshold: P=0.05/16,790).**
(TIF)

**S16 Fig. Cox proportional hazards model and cumulative incidence of atrial flutter/fibrillation by GWAS meta-height PRS quintiles in Han Taiwanese and Korean cohorts. Left panel:** Multivariate Cox model results for atrial flutter/fibrillation risk across height meta-analysis PRS quintiles, presenting hazard ratios (HRs) and 95% confidence intervals (CIs) relative to the reference group (PRS$_{40-60\%}$), adjusted for age, sex, and 10 principal components. **Right panel:** Kaplan–Meier survival curves illustrating the cumulative incidence of atrial flutter/fibrillation among individuals with low (PRS$_{0-20\%}$), middle (PRS$_{40-60\%}$), and high (PRS$_{80-100\%}$) genetic risk. *P*-values are based on log-rank tests comparing differences across PRS groups. The dashed line denotes the age at which each PRS group reaches a cumulative incidence of 10%. Numbers displayed below the curves represent participants at risk at each time point for the corresponding groups. Abbreviations: AF, atrial flutter/fibrillation; CI, confidence interval; HR, hazard ratio; PCs, principal components; PRS, polygenic risk score. (TIF)

**S1 Table. Basic characteristics of the 120,301 participants in the [TWB5] HEIGHT GWAS study.**
(XLSX)

**S2 Table. Genome-wide association study identified 293 lead single nucleotide polymorphisms across distinct genetic loci associated with body height in 120,301 Han Taiwanese individuals.**
(XLSX)

**S3 Table. GCTA-COJO joint/conditional results and pairwise LD (r²) between 293 TWB5 height lead SNPs and reported height index SNPs from the GWAS Catalog and ancestry panels (KoGES, BBJ, Yengo_EAS, EUR, Yengo_EUR), listing the nearest index SNP and its LD r² for each lead SNP.**
(XLSX)

**S4 Table. Colocalization analyses of 1-Mb genomic regions centered on 293 lead SNPs associated with body height, comparing TWB5 with other ancestry reference panels.**
(XLSX)

**S5 Table. SNP-based annotation of 1,185 body height susceptibility genes across 293 distinct loci in 120,301 Han Taiwanese individuals.**
(XLSX)

**S6 Table. Gene-based GWAS analysis identified 619 body height susceptibility genes in 120,301 Han Taiwanese individuals.**
(XLSX)

**S7 Table. Genetic correlation of [TWB5]HEIGHT with 130 diseases and traits of TWB database.**
(XLSX)

**S8 Table. Genetic correlation of [TWB5]HEIGHT with 111 diseases and traits of CMUH database.**
(XLSX)

**S9 Table. Genetic correlation of [TWB5]HEIGHT with 111 diseases and traits of CKB database.**
(XLSX)

**S10 Table. Genetic correlation of [TWB5]HEIGHT with 181 diseases and traits of BBJ database.**
(XLSX)

**S11 Table. Genetic correlation of [TWB5]HEIGHT with 71 diseases and traits of KoGES database.**
(XLSX)

**S12 Table. Basic characteristics of the FSS cases and controls in the [CMUH] FSS GWAS study.**
(XLSX)

**S13 Table. Genome-wide association study identified five height-reducing variants across distinct genetic loci associated with familial short stature in 2,050 cases and 27,966 controls of Han Taiwanese ancestry.**
(XLSX)

**S14 Table. GCTA-COJO joint/conditional results and pairwise LD (r²) between 5 familial short stature lead SNPs and reported height index SNPs from the GWAS Catalog and ancestry panels (KoGES, BBJ, Yengo_EAS, EUR, Yengo_EUR), listing the nearest index SNP and its LD r² for each lead SNP.**
(XLSX)

**S15 Table. Colocalization analyses of 1-Mb genomic regions centered on five lead SNPs associated with [CMUH] FSS, comparing CMUH with other ancestry reference panels for body height.**
(XLSX)

**S16 Table. SNP-based annotation of 26 susceptibility genes across five distinct loci associated with [CMUH]FSS in 2,050 cases and 27,966 controls of Han Taiwanese ancestry.**
(XLSX)

**S17 Table. Gene-based GWAS analysis identified 8 susceptibility genes associated with [CMUH]FSS in 2,050 cases and 27,966 controls of Han Taiwanese ancestry.**
(XLSX)

**S18 Table. Three stature gene lists used for Ingenuity Pathway Analysis (IPA).**
(XLSX)

**S19 Table. Ingenuity Pathway Analysis (IPA)-identified canonical signaling pathways associated with genes from [TWB5]HEIGHT, [TWB5+KoGES]HEIGHT, and [CMUH]FSS.**
(XLSX)

**S20 Table. Genetic correlation of [CMUH]FSS with 131 diseases and traits of TWB database.**
(XLSX)

**S21 Table. Genetic correlation of [CMUH]FSS with 111 diseases and traits of CMUH database.**
(XLSX)

**S22 Table. Genetic correlation of [CMUH]FSS with 118 diseases and traits of CKB database.**
(XLSX)

**S23 Table. Genetic correlation of [CMUH]FSS with 179 diseases and traits of BBJ database.**
(XLSX)

**S24 Table. Genetic correlation of [CMUH]FSS with 70 diseases and traits of KoGES database.**
(XLSX)

**S25 Table. Data quality and genetic correlations among body height and 31 key phenotypes in East Asians.**
(XLSX)

**S26 Table. Data quality and genetic correlations among familial short stature and 14 key phenotypes in East Asians.**
(XLSX)

**S27 Table. Basic characteristics of the independent CMUH Cohort (N=296,745) in the [TWB5]HEIGHT PRS-PheWAS study.**
(XLSX)

**S28 Table. Phenome-wide association study of height polygenic risk scores identified 13 disease phenotypes (*P*<0.05/1,090) in the China Medical University Hospital database.**
(XLSX)

**S29 Table. Basic characteristics of the independent CMUH Cohort (N=258,931) in the [CMUH]FSS PRS-PheWAS study.**
(XLSX)

**S30 Table. Phenome-wide association study of familial short stature polygenic risk scores identified 3 disease phenotypes (*P*<0.05/1,042) in the China Medical University Hospital database.**
(XLSX)

**S31 Table. Associations of stature- and GWAS-meta-analysis-derived polygenic risk scores with atrial fibrillation and endometriosis in Taiwanese and Japanese biobanks.**
(XLSX)

**S32 Table. Association of [TWB5] Height PRS with atrial flutter/fibrillation and endometriosis risk in the CMUH cohort based on univariate and multivariate logistic regression models.**
(XLSX)

**S33 Table. Body height and FSS instrumental variables for two-sample Mendelian randomization (MR) and multivariate MR analyses in East Asians.**
(XLSX)

**S34 Table. Causal effect of stature on endometriosis in East Asians: two-sample Mendelian randomization (MR) and multivariate MR (adjusted for age at menarche).**
(XLSX)

**S35 Table. Causal effect of stature on endometriosis in East Asians: two-sample Mendelian randomization (MR) and multivariate MR (adjusted for body weight).**
(XLSX)

**S36 Table. Body height instrumental variables for two-sample Mendelian randomization (MR) and multivariate MR analyses in East Asians.**
(XLSX)

**S37 Table. Causal effect of body height on atrial flutter/fibrillation using two-sample Mendelian randomization (MR) and multivariate MR (adjusted for age at menarche).**
(XLSX)

**S38 Table. Causal effect of body height on atrial flutter/fibrillation using two-sample Mendelian randomization (MR) and multivariate MR (adjusted for body weight).**
(XLSX)

**S39 Table. GWAS meta-analysis of Han Taiwanese and Korean cohorts identified 433 lead SNPs across distinct loci associated with body height.**
(XLSX)

**S40 Table. SNP-based annotation of 1,563 body height susceptibility genes across 433 distinct loci in Han Taiwanese and Korean cohorts.**
(XLSX)

**S41 Table. Gene-based GWAS analysis identified 967 body height susceptibility genes in Han Taiwanese and Korean cohorts.**
(XLSX)

**S1 File. STROBE Checklist.** Attribution: The STROBE checklist is reproduced from the STROBE Statement (Strengthening the Reporting of Observational Studies in Epidemiology) and is licensed under CC BY 4.0. Source: https://www.strobe-statement.org/.
(DOCX)

## Acknowledgments

We appreciate the Health Data Science Center, China Medical University Hospital (H107184 and H109315), and Taiwan Biobank (TWBR11005–09) for their administrative and technical support. We appreciate all contributors to BioBank Japan.

## Author contributions

**Conceptualization:** Ying-Ju Lin, Ting-Yuan Liu, Fuu-Jen Tsai.

**Formal analysis:** Ying-Ju Lin, Ting-Yuan Liu, Kuyuri Ariyoshi, Keiko Hikino, Chikashi Terao, Wen-Miin Liang, I-Ching Chou, Ting-Hsu Lin, Chiu-Chu Liao, Shao-Mei Huang, Fuu-Jen Tsai.

**Investigation:** Ying-Ju Lin, Ting-Yuan Liu, Jai-Sing Yang, Ju-Pi Li, Jian-Shiun Chiou, Hsing-Fang Lu, Chen-Hsing Chou, Fuu-Jen Tsai.

**Methodology:** Ying-Ju Lin, Ting-Yuan Liu, Jai-Sing Yang, Ju-Pi Li, Jian-Shiun Chiou, Hsing-Fang Lu, Kuyuri Ariyoshi, Keiko Hikino, Chikashi Terao, Chen-Hsing Chou, Wen-Miin Liang, I-Ching Chou, Ting-Hsu Lin, Chiu-Chu Liao, Shao-Mei Huang, Fuu-Jen Tsai.

**Writing – original draft:** Ying-Ju Lin, Ting-Yuan Liu, Fuu-Jen Tsai.

**Writing – review & editing:** Jai-Sing Yang, Ju-Pi Li, Jian-Shiun Chiou, Hsing-Fang Lu, Kuyuri Ariyoshi, Keiko Hikino, Chikashi Terao, Chen-Hsing Chou, Wen-Miin Liang, I-Ching Chou, Ting-Hsu Lin, Chiu-Chu Liao, Shao-Mei Huang.

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
