## [Decision Letter · Decision Letter 0]

22 Jul 2025

PGENETICS-D-25-00676

Unraveling the genetic links between stature and disease in East Asians: A multi-biobank genetic correlation and risk prediction study

PLOS Genetics

Dear Dr. Lin,

Thank you for submitting your manuscript to PLOS Genetics. After careful consideration, we feel that it has merit but does not fully meet PLOS Genetics's publication criteria as it currently stands. Therefore, we invite you to submit a revised version of the manuscript that addresses the points raised during the review process.

Please submit your revised manuscript within 90 days Sep 20 2025 11:59PM. If you will need more time than this to complete your revisions, please reply to this message or contact the journal office at plosgenetics@plos.org. Please include the following items when submitting your revised manuscript:

We look forward to receiving your revised manuscript.

Kind regards,

Themistocles L. Assimes

Academic Editor

PLOS Genetics

Anne O'Donnell-Luria

Section Editor

PLOS Genetics

Aimée Dudley

Editor-in-Chief

PLOS Genetics

Anne Goriely

Editor-in-Chief

PLOS Genetics

**Journal Requirements:**

At this stage, the following Authors/Authors require contributions: Ying-Ju Lin. Please ensure that the full contributions of each author are acknowledged in the "Add/Edit/Remove Authors" section of our submission form.

The list of CRediT author contributions may be found here: https://journals.plos.org/plosgenetics/s/authorship#loc-author-contributions

**Reviewers' comments:**

Reviewer's Responses to Questions

**Comments to the Authors:**

Reviewer #1: Lin et al. have conducted a GWAS analysis on height and FSS in a Han Taiwanese cohort. They further employed 5 East Asian Biobanks to access genetic correlations. They also performed a PheWAS and PRS analysis. They report genetic associations height and FSS and pleiotropic effects with AF and endometriosis. The results are not novel, as the reported loci are already reported before in larger GWAS studies and so are the PheWAS results for AF and endometriosis. The novelty of this manuscript is the genetic diversity, as the study is focusing on East Asian populations. However, the full potential of this study has not been realised here, as there is no attempt for a meta-analysis across the non-overlapping East-Asian Biobanks for available phenotypes. The results mainly focus on the CMUH cohort and only the genetic correlations consider the other biobanks individually. This results in underpowered conclusions. Furthermore, the methods are poorly described making the results difficult to interpret. In addition, tables and figures are not always informative and the result section is not expanded in enough detail. Discussion is also brief and some of the conclusions are not corroborated by the results.

Major points:

1. Please provide information on how phenotypes were measured/defined in the CMUH cohort, what was the age range and the length of follow-up. A supplementary table with the descriptives would be informative. Please also provide the ICD10 codes used to define FSS.

2. Please describe how lifestyle information was collected and quantified and the rationale behind residualizing the height with lifestyle adjustments before performing the GWAS. Please also explain whether the age adjustment refers to the age of recruitment or the age of the height measurement was used.

3. Please describe the PRS analysis and cox models in more detail.

4. Please describe how height and FSS loci were defined from the GWAS data and how the MAGMA analysis was performed.

5. It is unclear why a meta-analysis for available, non-overlapping biobanks was not attempted. The authors need to comment on that. Line 134 reports a total of 1.2M East Asians included in this study, which is quite misleading as the cohorts were analysed individually across this study.

6. Please discuss how the current GWAS in the CMUH cohort differs from your previous published GWAS for height in the same cohort: https://doi.org/10.1186/s12916-022-02450-w

7. Tables S1 and S3 are missing the effect and other allele, freq, n, beta and se stats. Please add.

8. Please explain the analysis used to derive the results included in Tables S2 and S4. Please also explain what the NAs are in that Tables; many of them reported for the “novel” genes. Please explain how novel genes were defined. Are these supposed to be causal genes? The methods for this analysis are missing.

9. Table S5 and S6: significance should be defined at a multiple testing not nominal significance. Several non-standard abbreviations are used in these tables but not explained.

10. Table S8: endometriosis should not be significant based on the Bonferroni threshold.

11. Figure 1 is misleading as it is not clear that the other 5 biobanks are ONLY used for the genetic correlation part.

12. Figure B and E: it is unclear how these plots were generated.

13. Figure 2C,F and Figure 3 seems to present the same results.

14. Figure 4: please make sure all text is readable, and the dots are replaced with up/down triangles to highlight the direction of effect.

15. Figure 5: A fairer comparison would be to set the middle PRS group as the ref. The right-hand side plots are not described in the results. What are the different age points where the curves diverge and are these significantly different?

16. In the Discussion, the authors claim that: “Herein, we identified 293 loci for body height and 5 for FSS in Han Taiwanese, revealing both overlapping and distinct genetic pathways.” The authors need to clearly state how many known and novel loci. In addition, there is no evidence of pathway analysis here, so the last part of this statement is not supported.

17. In the Discussion the authors claim that: “Our findings reveal unique genetic factors influencing stature in Han Taiwanese” but all of the GWAS hits were previously reported. How are there unique genetic factors?

18. In the Discussion the authors claim that: “Herein, GWAS, PheWAS, and PRS survival analyses were performed, revealing that genetically taller stature is linked to increased risks of AF and endometriosis, whereas shorter stature is associated with later onset and a reduced risk of endometriosis individuals”. This statement is not fully corroborated by the results, since the age of onset was not properly tested in the analysis. Furthermore, endometriosis was actually not Bonferroni significant with FSS.

19. Several key references reporting the association between low height and endometriosis are not included or discussed in this paper.

Reviewer #2: In “Unraveling the genetic links between stature and disease in East Asians: A multibiobank genetic correlation and risk prediction study”; the authors perform a wide-ranging analysis of the health risks associated with height, and additionally explore the risks associated with familial short stature, in previously under-studied east-asian populations. They perform GWAS, PheWAS and genetic correlation analyses and MAGMA gene analyses.

Overall, whilst I think the study concept is good, I think there are improvements to be made. Whilst I found the links to different health risks interesting, I would either like to see some analysis of other ancestries, to see if the association is unique or altered within the populations studied here, or a greater discussion on the literature for this issue.

I found it odd that none of the genetic loci were classified as new, but the MAGMA analysis highlighted new genes? This is particularly noticeable in ST3: you identify three single variants on chromosome 12, which you link to 20 separate genes in ST4. Can you justify this more clearly?

As an aside, it would be beneficial to check your criteria for defining a novel loci. If I understand correctly, you consider a loci as previously reported if there is a GWAS catalog hit within 1MB? I think this is quite conservative – could you do some co-localisation analysis? It may be that your loci is distinct to previous loci despite being proximal? Or your SNP is a better representation of the loci within EAS? Broadly, throughout the article, I also found the lack of discussion and comparison with the Yengo et al 2022 meta-analysis of 5M individuals for height quite jarring.

For the genetic correlations, I wonder whether a more targeted approach would be good, to lower your multiple-testing threshold?

I note that you found genetic correlation between height (and FSS) and endomeitriosis. However, as you state in the introduction, height is associated with menarche, which is further associated with reproductive health outcomes. I think more needs to be done to demonstrate the causal nature of this relationship, such as by adjusting for the genetic architecture of menarche.

Minor comments:

• Line 101 – I don’t think its accurate to say height growth ‘concludes at menarche’, as males don’t undergo menarche

• Line 101 – Yengo et al 2022 should to be discussed here

• Line 109 – reference for ‘most common form of short stature’

• Line 139 – I note you classify BMI as an FSS-specific trait. Recent articles (e.g. PMID 39818621) have discussed how ratio traits such as BMI are affected by both components of that ratio. It would be nice to see more justification overall of how you split traits into height/FSS specific groupings

• Line 152 & Figure 2 – Ideally I’d like to see the correlations across all the biobanks analysed for any trait which shows a significant correlation in one biobank

• Line 211 & 213 ‘lower risk of endometriosis’ – I would suggest rephrasing these statements throughout the manuscript to ‘protective effects’, to avoid confusion

Reviewer #3: This manuscript by Lin et al. provides a valuable contribution to our understanding of the genetic architecture of height, particularly in East Asian populations. The effort to integrate multiple large-scale biobanks across East Asia to explore the genetic underpinnings of stature and its potential links to disease is commendable. However, I don’t think the manuscript itself is well polished (as well as the figures), which makes it difficult to follow. The transparency of the analyses is another great concern. The manuscript requires substantial revision to improve its readability and methodological transparency. In addition to this, I have the following specific comments:

1. The authors conducted cross-trait genetic correlation analyses across five East Asian biobanks, selecting and categorizing a wide range of traits. I’m curious about the rationale and justification for this categorization (i.e., height specific, shared, and FSS specific). Additionally, I was unable to find a table listing all the phenotypes included in the genetic correlation analyses.

2. Fig 2A-B: The title of each plot is not clear to me, what do the numbers after the cohort names (e.g., TWB5, FSScontr27) represent?

3. Fig 2C,F: Here the P-value column refers to the value after multiple testing correction? And what method was used for this (FDR, Bonferroni?), I could not find it in the Methods section.

4. There is a lack of reporting the demographics for the population selected for the GWAS analysis.

5. The authors failed to provide some major details for each step of analysis in the Methods section. For instance, there is a lack of details on genotyping, imputation and quality control. For instance, what is the phenotypic variance explained by the identified loci? Have the authors calculated the genomic inflation?

6. There is a lack of functional follow-up in novel loci identified for the East Asian population.

Reviewer #4: The authors conducted new GWAS in the Han Taiwanese population for height and familial short stature, and explored genetic correlations with phenome-wide outcomes. They confirmed the previously established associations between tall stature and increased risk for atrial fibrillation (AF) and endometriosis. Furthermore, they identified that the height PRS was significantly associated with AF and endometriosis risk in the East Asian population. The analysis plan is solid, and the findings are robust. My comments are below.

Major comments

1. In clinical epidemiology, endometriosis is associated with early menarche and taller stature. However, it is well established that taller stature is generally associated with later menarche. This genetic analysis appears to recapitulate that pattern as well. This seems discrepant, as one would expect the pathway to be: early menarche, shorter stature, higher risk of endometriosis. Are there any explanations for this discrepancy? Could genetics contribute to this paradox?

2. The authors claim that the height PRS has utility in predicting AF and endometriosis. I was curious whether the height PRS captures genetic variance not explained by the AF-PRS or endometriosis PRS, which would support its additional value in predicting these two outcomes.

3. The authors reported genetic correlations between height and the phenome, as well as between FSS and the phenome. I was particularly interested in the degree of negative genetic correlation between height and FSS. Additionally, I was curious about the height effect sizes of the five loci associated with FSS.

4. In the abstract, the sentence "PheWAS further showed that tall stature increased risks of AF and endometriosis, while short stature reduced endometriosis risk." After reading through the manuscript, I understood that the authors tried to present that high height PRS is associated with increased risk for AF and endometriosis, but the PRS for FSS was associated with lower risk for endometriosis.

However, this is a bit difficult to understand at the first glance. As associations between high-height PRS and high endometriosis risk and high-FFS PRS and low endometriosis risk are indicating the same direction of associations, please reconsider the expression here.

Minor comments

5. Line 104: The authors refer to "the risk of short stature," but this phrasing may not be entirely neutral. Since short stature itself is not necessarily a pathological condition, and to avoid potential stigma, we recommend using more neutral language.

6. line 189 "A higher genetically determined short stature was linked to an increased risk of clinical short stature and reduced risks of endometriosis and atopic/contact dermatitis" is hard to interpret. Please reconsider rephrasing

7. Line 197 "Individuals in the highest PRS quintile showed significantly higher risks of AF (adjusted hazards ratio [HR]: 1.31, P = 6.76x10-12) and endometriosis (adjusted HR: 1.13, P = 1.45x10-3) and correspondingly higher cumulative incidence rates (AF: P < 0.0001; endometriosis:P = 0.0011) (Fig 5A)."

The authors report hazard ratios comparing the top and bottom quintiles of the PRS, which inflates the HR and limits interpretability. For example, this is akin to comparing disease risk between individuals under 20 and over 80 years old. A more interpretable estimate would compare the top 20% to the rest of the population. Also, reporting effect sizes by 1SD-PRS is helping interpretation and improve comparability of the results.

8. Line 214 "Overall, these findings highlighted the strong genetic associations between stature and major health outcomes in East Asians..." This might be overstating the utility of the study. While AF and endometriosis are certainly important outcomes, they do not represent major health outcomes as a whole.

9. Line 219 "Gene-based analysis using MAGMA identified 15 novel genes associated with body height (S2 Table)." I could not find the definition of "Novel" throughout the manuscript. Were not these genes identified in the large scale GWAS (ex. PMID: 36224396)? If so, how this novelty was determined?

10. Line 282 Please provide specific diagnosis code uses in this study.

11. PheWAS does not only indicate phenome-wide screening of associations for PRS. PheWAS includes pleiotropic scan of specific variants or phenome wide GWAS. Using more specific wording will help readers.

12. Line 343 "We conducted genome-wide association study (GWAS) analyses to obtain body mass index (BMI) and body fat percentage GWAS summary statistics from the Taiwan Biobank database" Is this relevant to this study? I also did not find description about data availability of GWAS for Height and FSS conducted in this study.

**Have all data underlying the figures and results presented in the manuscript been provided?**

Reviewer #1: **No:** Complete PheWAS results not provided as supplementary table, only significant results. CMUH GWAS not deposited but available upon request from corresponding author.

Reviewer #2: Yes

Reviewer #3: Yes

Reviewer #4: Yes

PLOS authors have the option to publish the peer review history of their article (what does this mean? ). If published, this will include your full peer review and any attached files.

**Do you want your identity to be public for this peer review?** For information about this choice, including consent withdrawal, please see our Privacy Policy .

Reviewer #1: No

Reviewer #2: No

Reviewer #3: No

Reviewer #4: **Yes:** Satoshi Koyama

**Figure resubmission:**
---

## [Decision Letter · Decision Letter 1]

17 Nov 2025

PGENETICS-D-25-00676R1

Unraveling the genetic links between stature and disease in East Asians: A multi-biobank genetic correlation and risk prediction study

PLOS Genetics

Dear Dr. Lin,

Thank you for submitting your manuscript to PLOS Genetics. Two out of the four reviewers were very satisfied with your response and updates to your manuscript. A third requested a minor revision, and the fourth reviewer asked for further clarification, not only on some of the original analyses but also on some of the new analyses. After careful consideration, we feel that it has merit but does not fully meet PLOS Genetics's publication criteria as it currently stands. Therefore, we invite you to submit a revised version of the manuscript that addresses the points raised by the two reviewers.

We look forward to receiving your revised manuscript.

Kind regards,

Themistocles L. Assimes

Academic Editor

PLOS Genetics

Anne O'Donnell-Luria

Section Editor

PLOS Genetics

Aimée Dudley

Editor-in-Chief

PLOS Genetics

Anne Goriely

Editor-in-Chief

PLOS Genetics

**Journal Requirements:**

1) Please provide an Author Summary. This should appear in your manuscript between the Abstract (if applicable) and the Introduction, and should be 150-200 words long. The aim should be to make your findings accessible to a wide audience that includes both scientists and non-scientists. Sample summaries can be found on our website under Submission Guidelines:

https://journals.plos.org/plosgenetics/s/submission-guidelines#loc-parts-of-a-submission

2) The manuscript PDF must be well-organized and in good condition for peer review purposes. It should be a clean file, free of highlighting or tracked changes.

**Reviewers' comments:**

Reviewer's Responses to Questions

**Comments to the Authors:**

Reviewer #1: Lin et al. have submitted a revised version of their manuscript, with several additional analysis added. Some of my previous comments have been addressed but not all and the new additions have generated more concerns for me.

One major comment is that currently the supplementary tables in word are impossible to consider in a meaningful way. The authors should provide all supplementary tables in an excel format.

In addition:

1. The authors have removed the sentence “Herein, GWAS, PheWAS, and PRS survival analyses were performed, revealing that genetically taller stature is linked to increased risks of AF and endometriosis, whereas shorter stature is associated with later onset and a reduced risk of endometriosis individuals”, since they age of onset was not properly tested. But they still mention in the abstract and the discussion that "Higher height PRS further predicted an earlier and higher AF incidence, consistently replicated in survival analyses of meta-analysis–derived PRSs". It is not clear from their analysis, how the "earlier AF incidence" is justified.

2. The locus definition is still unclear to me; there is no description as to what a "distinct" locus means in terms of distance.

3. The classification of novel vs known loci is very confusing. Standard practice is to check all published variants for the trait of interest (from the GWAS catalog) and check via LD if current variants are proxies or not of published variants. I am unsure as to why the authors have performed a co-localization analysis, as this is more suitable for identifying variants that affect more than one traits. If they wish to identify independent/novel variants with published loci, them they need to perform conditional analysis not co-localization.

4. Lines 173-176 are more suitable int he Methods section.

5. Based on the FSS description, it becomes apparent that only children are included in the cases (since the ICD codes only refer the children conditions), while all the controls are adults. This raises major concerns about the suitability of the control group and the selection bias introduced here.

6. It is unclear if the FUMA and MAGMA analysis are meant to identify causative genes but seems counter-intuitive that 5 variants are mapped to 27 genes with no prioritization.

7. Line 195 implies causality that is not supported by the type of the analyses presented in that section. Same for line 204.

8. PheWAS results need to be corrected for multiple testing (see line 220).

9. Line 223 mentions an "independent" cohort but there is no information of that cohort anywhere.

10. There seems to be lack of independent studies for validating the PRSs. The same cohort is being used to derive GWAS effect sizes, train, test and validate the PRS. This will overestimate the PRS performance.

11. The authors seem to use an AF PRS and an endometriosis PRS but there is no information on how these PRS were derived.

12. The authors have performed a series of MR analysis but there is no description of this in the methods. In addition, it would be helpful if the results presented in figure S8 were also included in a supplementary table.

Reviewer #2: The authors have substantially improved their manuscript and addressed all of my concerns.

Reviewer #3: Lin and colleagues have significantly improved the overall transparency of their work; the readability has also been significantly improved. I am happy with authors’ response regarding points I previously raised.

One major point I find is that, I can see that the authors had added the part using Mendelian randomization to examine potential causal associations, but I did not find the detailed methodology for MR in the Methods section. Particularly, how instruments were selected in both univariable MR and multivariable MR; if the three core assumptions for MR were tested (I can see that the authors have adopted MR-PRESSO, but only from the supplementary materials); and which R package was used for these analyses. MVMR estimates could be biased if the conditional F-statistics are low (instrument strength). These details should be provided.

Minor point:

I was trying to investigate some of the supplementary tables, but tables in a word document make them very difficult to navigate, compared to a spreadsheet file like excel. I’m not sure if this is the requirement from the journal.

Reviewer #4: Thank you for the comprehensive revisions to the manuscript and the inclusion of additional interesting findings. I have no further comments.

**Have all data underlying the figures and results presented in the manuscript been provided?**

Reviewer #1: Yes

Reviewer #2: Yes

Reviewer #3: Yes

Reviewer #4: Yes

PLOS authors have the option to publish the peer review history of their article (what does this mean? ). If published, this will include your full peer review and any attached files.

**Do you want your identity to be public for this peer review?** For information about this choice, including consent withdrawal, please see our Privacy Policy .

Reviewer #1: No

Reviewer #2: No

Reviewer #3: No

Reviewer #4: **Yes:** Satoshi Koyama

**Figure resubmission:**
---

## [Decision Letter · Decision Letter 2]

13 Jan 2026

Dear Dr Lin,

We are pleased to inform you that your manuscript entitled "Unraveling the genetic links between stature and disease in East Asians: A multi-biobank genetic correlation and risk prediction study" has been editorially accepted for publication in PLOS Genetics. Congratulations!

Yours sincerely,

Themistocles L. Assimes

Academic Editor

PLOS Genetics

Anne O'Donnell-Luria

Section Editor

PLOS Genetics

Aimée Dudley

Editor-in-Chief

PLOS Genetics

Anne Goriely

Editor-in-Chief

PLOS Genetics

BlueSky: @plos.bsky.social

Comments from the reviewers (if applicable):

Reviewer's Responses to Questions

**Comments to the Authors:**

Reviewer #1: The authors have addressed my previous comments. I am happy with the revised manuscript.

Reviewer #3: I appreciate the authors’ efforts in clarifying the Mendelian randomization methodology in their revised manuscript, and the supplemental tables are now provided in Excel format, making them much easier to examine. I have no further comments on this manuscript.

**Have all data underlying the figures and results presented in the manuscript been provided?**

Reviewer #1: Yes

Reviewer #3: None

PLOS authors have the option to publish the peer review history of their article (what does this mean? ). If published, this will include your full peer review and any attached files.

**Do you want your identity to be public for this peer review?** For information about this choice, including consent withdrawal, please see our Privacy Policy .

Reviewer #1: No

Reviewer #3: No

**Data Deposition**

http://datadryad.org/submit?journalID=pgenetics&manu=PGENETICS-D-25-00676R2

**Press Queries**

---

## [Editor Report · Acceptance letter]

PGENETICS-D-25-00676R2

Unraveling the genetic links between stature and disease in East Asians: A multi-biobank genetic correlation and risk prediction study

Dear Dr Lin,

We are pleased to inform you that your manuscript entitled "

Unraveling the genetic links between stature and disease in East Asians: A multi-biobank genetic correlation and risk prediction study" has been formally accepted for publication in PLOS Genetics! Your manuscript is now with our production department and you will be notified of the publication date in due course.

With kind regards,

Anita Estes

PLOS Genetics

On behalf of:
